# Accelerated Variance Reduced Stochastic Extragradient Method for Sparse Machine Learning Problems

## Abstract

Recently, many stochastic gradient descent algorithms with variance reduction have been proposed. Moreover, their proximal variants such as Prox-SVRG can effectively solve non-smooth problems, which makes that they are widely applied in many machine learning problems. However, the introduction of proximal operator will result in the error of the optimal value. In order to address this issue, we introduce the idea of extragradient and propose a novel accelerated variance reduced stochastic extragradient descent (AVR-SExtraGD) algorithm, which inherits the advantages of Prox-SVRG and momentum acceleration techniques. Moreover, our theoretical analysis shows that AVR-SExtraGD enjoys the best-known convergence rates and oracle complexities of stochastic first-order algorithms such as Katyusha for both strongly convex and non-strongly convex problems. Finally, our experimental results show that for ERM problems and robust face recognition via sparse representation, our AVR-SExtraGD can yield better performance than state-of-the-art algorithms such as Prox-SVRG and Katyusha. The asynchronous variant of AVR-SExtraGD outperforms KroMagnon and ASAGA, which are the asynchronous variants of SVRG and SAGA, respectively.

## 1 Introduction

In this paper, we mainly consider the following composite convex optimization problem:

$$\min_{x \in \mathbb{R}^d} \left\{ P(x) \stackrel{def}{=} F(x) + R(x) = \frac{1}{n} \sum_{i=1}^{n} f_i(x) + R(x) \right\} \tag{1}$$

where $F(x) : \mathbb{R}^d \to \mathbb{R}$ is the average of smooth convex component functions $f_i(x)$, and $R(x)$ is a relatively simple convex function (but may not be differentiable). In this paper, we use $\|\cdot\|$ to denote the standard Euclidean norm, and $\|\cdot\|_1$ to denote the $\ell_1$-norm. Moreover, we use $P_*$ to denote the real optimal value of $P(\cdot)$, and $\widehat{P}_*$ to denote the optimal value obtained by algorithms.

This form of optimization problems often appears in machine learning, signal processing, data science, statistics and operations research, and has a wide range of applications such as regularized empirical risk minimization (ERM), sparse coding for image and video recovery, and representation learning for object recognition. Specifically, for a collection of given training examples $\{(a_1, b_1), ..., (a_n, b_n)\}$, where $a_i \in \mathbb{R}^d$, $b_i \in \mathbb{R}$ $(i = 1, 2, ..., n)$ and $a_i$ is a feature vector, while $b_i$ is the desired response. When $f_i(x) = \frac{1}{2}(a_i^T x - b_i)^2$, we can obtain the ridge regression problem by setting $R(x) = \frac{\lambda}{2}\|x\|^2$. We also get the Lasso or Elastic-Net problems by setting $R(x) = \lambda\|x\|_1$ or $R(x) = \frac{\lambda_2}{2}\|x\|^2 + \lambda_1\|x\|_1$, respectively. Moreover, if we set $f_i(x) = \log(1 + \exp(-b_i x^T a_i))$, we will get the regularized logistic regression problem.

### 1.1 Recent Research Progress

The proximal gradient descent (PGD) method is a standard and effective method for Problem (1), and can achieve linear convergence for strongly convex problems. Its accelerated algorithms, e.g., accelerated proximal gradient (APG) (Tseng (2008); Beck & Teboulle (2009)), attain the convergence rate of $\mathcal{O}(1/T^2)$ for non-strongly convex problems, where $T$ denotes the number of iterations.

In recent years, stochastic gradient descent (SGD) has been successfully applied to many large-scale learning problems, such as training for deep networks and linear prediction (Tong (2004)), because of its significantly lower per-iteration complexity than deterministic methods, i.e., $\mathcal{O}(d)$ vs. $\mathcal{O}(nd)$. Besides, many tricks for SGD have also been proposed, such as Loshchilov & Hutter (2016). However, the variance of the stochastic gradient may be large due to random sampling (Johnson & Tong (2013)), which leads that the algorithm requires a gradually reduced step size, thus it will converge slow. Even under the strongly convex condition, SGD only achieves a sub-linear convergence rate $\mathcal{O}(1/T)$. Recently, many SGD methods with variance reduction have been proposed. For the case of $R(x) = 0$, Roux et al. (2012) developed a *stochastic average gradient descent* (SAG) method, which is a randomized variant of the *incremental aggregated gradient* method proposed by Blatt et al. (2007). Then *stochastic variance reduced gradient* (SVRG) (Johnson & Tong (2013)) was proposed, and has been widely introduced into various subsequent optimization algorithms, due to its lower storage space (i.e., $O(d)$) than that of SAG (i.e., $O(nd)$). SVRG reduced the variance effectively by changing the estimation of stochastic gradients. The introduction of a snapshot point $\tilde{x}$ mainly has the effect of correcting the direction of gradient descent, and reduces the variance. Later, Konečný & Richtárik (2013) proposed the semi-stochastic gradient descent methods as well as their mini-batch version (Konečný et al. (2014)). And their asynchronous distributed variant (Ruiliang et al. (2016)) is also been proposed later. More recently, Lin & Tong (2014) proposed the Prox-SVRG method, which introduced the proximal operator, and then applied the idea of SVRG to solve the non-smooth optimization problems. However, Prox-SVRG can only be used to solve the strongly convex optimization problems. In order to solve the non-strongly convex problems, Zeyuan & Yuan (2016) proposed the SVRG++ algorithm. Besides, to accelerate the algorithm and reducing the complexity, by combining the main ideas of APG and Prox-SVRG, Nitanda (2014) proposed an *accelerated variance reduction proximal stochastic gradient descent* (Acc-Prox-SVRG) method, which can effectively reduce the complexity of the algorithm compared to the two basic algorithms. Very recently, Zeyuan (2017) developed a novel *Katyusha* algorithm which introduced the Katyusha momentum to accelerate the algorithm. With the development of parallel and distributed computing which can effectively reduce computing time and improve performance, Ryu & Wotao (2017) came up with an algorithm called *Proximal Proximal Gradient*, which combined the proximal gradient method and ADMM (Gabay & Mercier (1976)). Furthermore, it is easy to implement in parallel and distributed environments because of its innovative algorithm structure.

## 1.2 Our Main Contributions

We find that due to the introduction of proximal operator, there is a gap between $\widehat{P}_*$ and $P_*$, and its theoretical derivation can be seen in Appendix A. To address this issue, Nguyen et al. (2017) proposed the idea of extragradient which can be seen as a guide during the process, and introduced it into the optimization problems. Intuitively, this additional iteration allows us to examine the geometry of the problem and consider its curvature information, which is one of the most important bottlenecks for first order methods. By using the idea of extragradient, we can get a better result in each inner-iteration. Therefore, the idea of extragradient is our main motivation. In this paper, we propose a novel algorithm for solving non-smooth optimization problems. The main contributions of this paper are summarized as follows.

● In order to improve the result of the gap between $\widehat{P}_*$ and $P_*$, and achieve fast convergence, a novel algorithm, which combines the idea of extragradient, Prox-SVRG and the trick of momentum acceleration, is proposed, called *accelerated variance reduced stochastic extragradient descent* (AVR-SExtraGD).

● We provide the convergence analysis of our algorithm, which shows that AVR-SExtraGD achieves linear convergence for strongly convex problems, and the convergence condition in the non-strongly convex case is also given. According to the convergence rate, we know that AVR-SExtraGD has the same excellent result as the best-known algorithms, such as Katyusha.

● Finally, we show by experiments that the performance of AVR-SExtraGD (as well as VR-SExtraGD, which is the basic algorithm of AVR-SExtraGD) is obviously better than the popular algorithm, Prox-SVRG, which confirms the advantage of extragradient. For the widely used accelerated algorithm, Katyusha, the performance of our algorithm is still improved.

## 2 RELATED WORK

### 2.1 BASIC ASSUMPTIONS

We first make the following assumptions to solve the problem (1):

**Assumption 1** (Smoothness). *The convex function $F(\cdot)$ is L-smooth, i.e., there exists a constant $L > 0$ such that for any $x, y \in \mathbb{R}^d$, $\|\nabla F(x) - \nabla F(y)\| \leq L\|x - y\|$.*

**Assumption 2** (Lower Semi-continuity). *The regularization function $R(\cdot)$ is a lower semi-continuous function, i.e., $\forall x_0 \in \mathbb{R}^d$,*

$$\liminf_{x \to x_0} R(x) \geq R(x_0).$$

*But it is not necessarily differentiable or continuous.*

**Assumption 3** (Strong Convexity). *In Problem (1), the function $R(\cdot)$ is $\mu$-strongly convex, i.e., there exists a constant $\mu > 0$ such that for all $x, y \in \mathbb{R}^d$, it holds that*

$$R(x) \geq R(y) + \langle \mathcal{G}, \, x - y \rangle + \frac{\mu}{2}\|x - y\|^2, \tag{2}$$

*where $\mathcal{G} \in \partial R(y)$ which is the set of sub-gradient of $R(\cdot)$ at $y$.*

### 2.2 PROX-SVRG AND EXTRAGRADIENT DESCENT METHODS

An effective method for solving Problem (1) is Prox-SVRG which improved Prox-FG (Lions & Mercier (1979)) and Prox-SG (Langford et al. (2009)) by introducing the stochastic gradient and combining the idea of SVRG, respectively. For strongly convex problems, Prox-SVRG can reach linear convergence with a constant step size, and its main update rules are

$$\tilde{\nabla} f_{i_k}(x_{k-1}) = \nabla f_{i_k}(x_{k-1}) - \nabla f_{i_k}(\tilde{x}) + \nabla F(\tilde{x}); \;\; x_k = \text{Prox}_\eta^R(x_{k-1} - \eta \tilde{\nabla} f_{i_k}(x_{k-1})), \tag{3}$$

where $\tilde{x}$ is the snapshot point used in SVRG, $\tilde{\nabla} f_{i_k}(x_{k-1})$ is the variance reduced stochastic gradient estimator, and $\text{Prox}_\eta^R(\cdot)$ is the proximal operator. Although Prox-SVRG can converge fast, because of proximal operator, the final solution has the deviation, which makes the solution inaccurate, thus Prox-SVRG still needs to be further improved, which is our important motivation.

The extragradient method was first proposed by Korpelevič (1976). It is a classical method for solving variational inequality problems, and it generates an estimation sequence by using two projection gradients in each iteration. By combining this idea with some first-order descent methods, Nguyen et al. (2017) proposed an extended extragradient method (EEG) which can effectively solve the problem (1), and can also solve relatively more general problems as follows:

$$\min_{x \in \mathbb{R}^d} \left\{ P(x) \overset{def}{=} F(x) + R(x) \right\}$$

where $F(x)$ is not necessarily composed by multiple functions $f_i(x)$. Unlike the classical extragradient method, EEG uses proximal gradient instead of orthogonal projection in each iteration. The main update rules of EEG are

$$y_k = \text{Prox}_{s_k}^R(x_k - s_k \nabla F(x_k)); \;\; x_{k+1} = \text{Prox}_{\alpha_k}^R(x_k - \alpha_k \nabla F(y_k)),$$

where $s_k$ and $\alpha_k$ are two step sizes. From the update rules of EEG, we can see that in each iteration, EEG needs to calculate two gradients, which will definitely slow down the algorithm. Therefore, the algorithm needs to be further accelerated by an efficient technique.

### 2.3 MOMENTUM ACCELERATION AND MIG

Firstly, we introduce the momentum acceleration technique whose main update rules are

$$v_{dw_t} = \beta v_{dw_{t-1}} + (1 - \beta)dw_t; \;\; w_t = w_{t-1} - \alpha v_{dw_t},$$

where $dw$ is the gradient of the objective function at $w$, $\beta$ is a parameter, and $\alpha$ is a step size. The update rules take not only the gradient of the current position, but also the gradient of the past position into account, which makes the final descent direction of $w_t$ after using momentum

---

**Algorithm 1** AVR-SExtraGD

---

**Input:** Initial vector $x_0$, the number of epochs $S$, the number of iterations $m$ per epoch, the step sizes $\eta_1, \eta_2$, momentum parameter $\beta$, and the set $K$.
**Initialize:** $\tilde{x}^0 = x_0^1 = x_0$, $\rho = 1 + \eta\mu$.
1: **for** $s = 1, 2, \ldots, S$ **do**
2:     Compute $\nabla F(\tilde{x}^{s-1})$;
3:     $\beta_s = \beta$ (SC) or $\beta_s = \frac{2}{s+4}$ (non-SC);
4:     **for** $k = 1, 2, \ldots, m$ **do**
5:         Pick $i_k$ uniformly at random from $\{1, ..., n\}$;
6:         **if** $k \in K$ **then**
7:             $x_{k-1/2}^s = \mathrm{Prox}_{\eta_1}^R \left( x_{k-1}^s - \eta_1 \tilde{\nabla} f_{i_k}(\beta_s x_{k-1}^s + (1-\beta_s)\tilde{x}^{s-1}) \right)$;
8:             $x_k^s = \mathrm{Prox}_{\eta_2}^R \left( x_{k-1/2}^s - \eta_2 \tilde{\nabla} f_{i_k}(\beta_s x_{k-1/2}^s + (1-\beta_s)\tilde{x}^{s-1}) \right)$;
9:         **else**
10:            $x_k^s = \mathrm{Prox}_{\eta_1}^R (x_{k-1}^s - \eta_1 \tilde{\nabla} f_{i_k}(\beta_s x_{k-1}^s + (1-\beta_s)\tilde{x}^{s-1}))$;
11:         **end if**
12:     **end for**
13:     $\tilde{x}^s = \beta_s (\sum_{k=1}^m \rho^{k-1})^{-1} \sum_{k=1}^m \rho^{k-1} \frac{x_{k-1/2}^s + x_k^s}{2} + (1-\beta_s)\tilde{x}^{s-1}$ (SC)
       or $\tilde{x}^s = \frac{\beta_s}{m} \sum_{k=1}^m \frac{x_{k-1/2}^s + x_k^s}{2} + (1-\beta_s)\tilde{x}^{s-1}$ (non-SC);
14:     $x_0^{s+1} = x_m^s$;
15: **end for**
**Output:** $\tilde{x}^S$.

---

reduce the oscillation of descent, thus this method can effectively accelerate the convergence of the algorithm.

According to the Nesterov's momentum, lots of accelerated algorithms were proposed, such as APG and Acc-Prox-SVRG. Later, Zeyuan (2017) proposed Katyusha to further accelerate the algorithm, and MiG (Kaiwen et al. (2018)) was proposed to simplify the structure of Katyusha, and the momentum acceleration of MiG is embodied in each iteration as follows:

$$y_{k-1}^s = \beta_s x_{k-1}^s + (1 - \beta_s)\tilde{x}_{s-1}.$$

Moreover, it is easy to get that the oracle complexity of MiG is less than that of Prox-SVRG and APG, which means that MiG can effectively accelerate the original Prox-SVRG algorithm. Therefore, we can also use this acceleration technique to accelerate our algorithm and address the issue of slow convergence due to the calculations of two different gradients.

## 3   Our AVR-SExtraGD Method

We note that EEG requires computing two full gradients in each iteration, which will take a lot of time for large-scale machine learning problems. Therefore, we first consider and propose the stochastic variant of the algorithm, namely *stochastic extragradient descent* (SExtraGD), to reduce the per-iteration computational complexity, and further propose an efficient *variance reduced stochastic extragradient descent* (VR-SExtraGD) algorithm. Their main update rules and the detailed algorithm of VR-SExtraGD can be found in Appendix C.

On the basis of VR-SExtraGD, we refer to the momentum acceleration technique proposed in MiG, and propose an innovative accelerated variance reduced stochastic extragradient descent algorithm, called **AVR-SExtraGD**. It is used to solve non-smooth (both SC and non-SC) optimization problems. To further accelerate the algorithm and address the issue of slow convergence speed caused by two gradients in each inner-iteration, only part of the iterations are updated by extragradient descent. Our AVR-SExtraGD algorithm is outlined in **Algorithm 1**.

Firstly, we give some explanation for Algorithm 1. For our AVR-SExtraGD, we need to compute a full gradient of $F(x)$. And Step 3 in Algorithm 1 is the selection of momentum parameter. Step 6 to Step 8 are the update rule of AVR-SExtraGD, and Step 9 is the update rule of MiG. Here we only use AVR-SExtraGD in the set $K$. Step 13 is the formulation of the snapshot point. Finally, we

give the set of the start point of next inner-iteration in Step 14. Moreover, our output is the snapshot point of the last outer iteration.

And for our AVR-SExtraGD algorithm, we have the following remarks.

● Following the requirement of step sizes in EEG, the step sizes in our algorithms also need to satisfy similar conditions. After combining all the conditions, we get the conditions: $\eta_1 \leq \frac{1}{2L}$, $\eta_2 \leq \frac{1}{L} - \eta_1$.

● In AVR-SExtraGD, we use one more trick to speed up the algorithm, that is, only part of the iterations (i.e., when $k \in K$) are updated by extragradient descent, and the rest of the iterations are still updated by the update rules of MiG. For different problems and different data sets, we manually adjust the choice of $K$, and the details can be seen in Section 5.3.

● For the momentum parameter $\beta$, when $P(\cdot)$ is a strongly convex function, we can set $\beta$ as a constant which is generally set as 0.9. And $\beta$ is also set as $\frac{1-\sqrt{\mu\eta_2}}{1+\sqrt{\mu\eta_2}}$ in Acc-Prox-SVRG, while we set $\beta = 0.9$ in our AVR-SExtraGD. However, when $P(\cdot)$ is non-strongly convex, the value of $\beta$ in each iteration is no longer fixed. We set $\beta_s$ as a decreasing sequence, which satisfies $\frac{1}{\beta_{s-1}^2} \geq \frac{1-\beta_s}{\beta_s^2}$. Particularly, in AVR-SExtraGD, we set $\beta_s = \frac{2}{s+4}$, which satisfies the inequality defined above.

As we all know, for the general GD method, the iterate $x_k$ in each iteration eventually converges to the real optimal point of the function, so there is no error in the final optimal value. Therefore, the proximal operator will introduce a bad result in convergence.

To adress this issue, we introduce the idea of extragradient, which takes one more proximal operator than Prox-SVRG. And according to the idea of extragradient, we know that the update structure of EEG can make use of the curvature information of the objective function. Although we change the original EEG into a stochastic version, the advantage of the extragradient structure is still retained to some degree, and thus our algorithm can get a better result than the algorithm without extragradient.That is, although the method of extragradient can not directly reduce the gap of the optimal value, it can improve the bad result brought by the gap, and obtain a better result after every inner loop, Thus, AVR-SExtraGD can improve the accuracy of the algorithm.

In summary, our AVR-SExtraGD method combines the advantage of Prox-SG for solving non-smooth optimization problems, the advantage of EEG, and the trick of SVRG to reduce the variance of stochastic gradient. And it is further accelerated by introducing the momentum acceleration used in MiG. Therefore, our algorithm has more advantages than the basic algorithms mentioned above.

## 4 CONVERGENCE ANALYSIS

In this section, we analyze the convergence properties of AVR-SExtraGD under strongly convex and non-strongly convex conditions. For convenience analysis, we use $\tilde{\nabla}_{i_k} F(\cdot)$ to denote $\tilde{\nabla} f_{i_k}(\cdot)$, that defined in (3) in the analysis of AVR-SExtraGD. We give some key lemmas, which are important to prove the convergence of AVR-SExtraGD in Appendix B, and all the proofs of our lemmas and theorems in this section are also given in Appendix B.

### 4.1 FOR SC PROBLEMS

For strongly convex problems, the linear convergence of AVR-SExtraGD can be guaranteed by the following theorem.

**Theorem 1** (Strongly Convex). *Suppose that Assumptions 1, 2 and 3 hold, and let* $x_* = \arg\min_x P(x)$. *In addition, assume* $\eta_1 = \eta_2 = \eta > 0$ *and* $L\beta + \frac{L\beta}{1-\beta} \leq \frac{1}{\eta}$. *Then, by appropriately choosing* $\eta, \beta$ *and* $m = \Theta(n)$, *Algorithm 1 achieves an* $\epsilon$-*additive error with following oracle complexities in expectation:*

$$\begin{cases} \mathcal{O}(\sqrt{\kappa n} \log \frac{P(x_0)-P(x_*)}{\epsilon}), & \text{if } \frac{m}{\kappa} \leq \frac{3}{4}, \\ \mathcal{O}(n \log \frac{P(x_0)-P(x_*)}{\epsilon}), & \text{if } \frac{m}{\kappa} > \frac{3}{4}, \end{cases}$$

*which also means that for SC problems, the oracle complexity of Algorithm 1 is* $\mathcal{O}((n + \sqrt{\kappa n}) \log \frac{P(x_0)-P(x_*)}{\epsilon})$.

This result means that for strongly convex problems, AVR-SExtraGD achieves linear convergence and enjoys the best-known oracle complexity of stochastic first-order algorithms, such as Katyusha.

## 4.2 FOR NON-SC PROBLEMS

The convergence of AVR-SExtraGD for solving non-SC problems can be guaranteed by the following theorem.

**Theorem 2** (Non-Strongly Convex). *Suppose that Assumptions 1 and 2 hold, and let $x_* = \arg\min_x P(x)$. In addition, assume $\eta_1 = \eta_2 = \eta = \frac{1}{L\alpha} > 0$ and $1 - \beta_s - \frac{1}{\alpha-1} \geq 0$, where $\alpha$ is a constant. Then by setting $\beta_s = \frac{2}{s+4}$, we have*

$$\mathbb{E}[P(\tilde{x}^S) - P(x_*)] \leq \frac{4(1-\beta_1)}{(S+4)^2 \beta_1^2}(P(x_0) - P(x_*)) + \frac{L\alpha}{(S+4)^2 m}\|x_0 - x_*\|^2,$$

*which also means that when we choose $m = \Theta(n)$, Algorithm 1 achieves the following oracle complexity in expectation:*

$$\mathcal{O}(n\sqrt{\frac{P(x_0) - P(x_*)}{\epsilon}} + \sqrt{\frac{nL\|x_0 - x_*\|^2}{\epsilon}}).$$

The result shows that AVR-SExtraGD enjoys the same oracle complexity as Katyusha and MiG, which is close to the best-known complexity in this case (i.e., $\mathcal{O}(n\log\frac{1}{\epsilon} + \sqrt{\frac{nL}{\epsilon}})$). In addition, we also analyze the convergence of VR-SExtraGD and give and prove the related lemmas and theorems to guarantee its convergence, which can be found in Appendix D.

## 5 EXPERIMENTS

In this section, we evaluate the performance of AVR-SExtraGD and compare it with its counterparts including Prox-SVRG and Katyusha on real-world data sets, whose information is shown as Table 2 in Appendix E.

Besides, for these real-world data sets, we consider the two common problem models: Lasso and Elastic-Net. We also apply our algorithm to face recognition tasks and compare it with the compared algorithms. Next, we give the setup of the related parameters as follows:

• **Regularization Parameters:** The regularization parameters for real-world datasets are shown in Table 2.

• **The Number of Inner-Iteration:** The number of inner-iterations of Katyusha and Prox-SVRG is usually set as $m = 2n$. Our algorithm adds one more gradient calculation in each inner-iteration than Prox-SVRG, and for an equal complexity of each epoch, we set $m = n$ in AVR-SExtraGD, so that in each epoch, all the three algorithms require calculating $3n$ stochastic gradients. What's more, the reasonableness of such a setting can be found in Sebbouh et al. (2019).

• **Step Sizes:** We set our step sizes as: $\eta_1 = \frac{2}{5L}, \eta_2 = \frac{3}{5L}$. We note that the selected step sizes do not satisfy the conditions requested in the remark of Section 3. Nevertheless, we can see from the experimental results that our algorithm still converges well, which means that in practice experiments, we can choose larger step sizes to improve the convergence speed.

For fair comparison, we implemented all the methods in C++ with a Matlab interface, and performed all the experiments on a PC with an Intel i7-7700 CPU and 32GB RAM.

## 5.1 RESULTS OF LASSO, ELASTIC-NET AND LOGISTIC REGRESSION

In this part, we consider three common problems, including Lasso, Elastic-Net and the $\ell_1$-norm regularized logistic regression, whose models can be found in Appendix E.

Figure 1 shows the performance of all the algorithms for Lasso and Elastic-Net on all the data sets. For running time, our AVR-SExtraGD obviously outperforms Prox-SVRG and Kayusha, which shows the faster convergence speed of AVR-SExtraGD than Katyusha, and justifies that the extra-gradient and the momentum acceleration are able to improve Prox-SVRG efficiently. Moreover, for

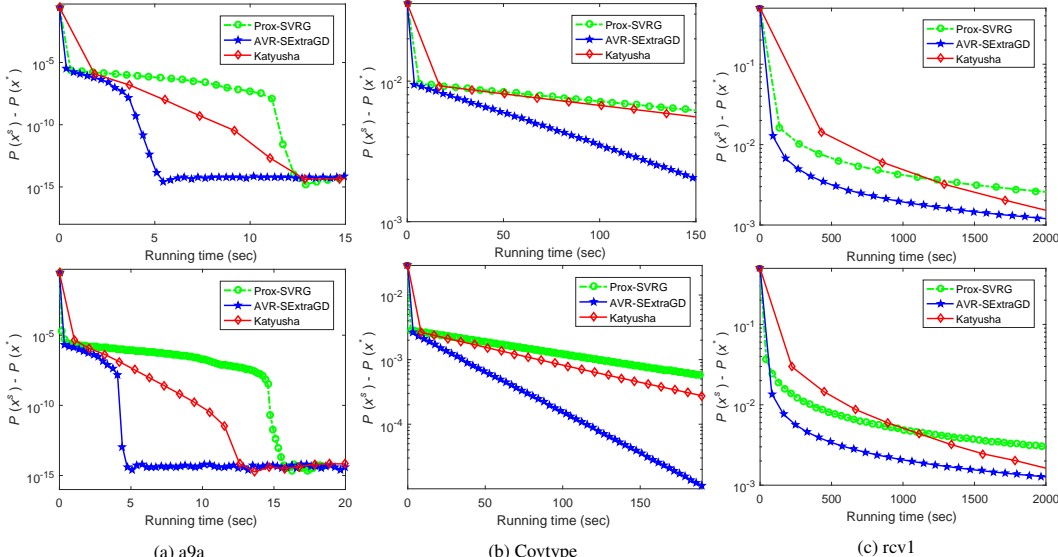

Figure 1: Comparison of experimental results of different algorithms for Lasso (top) and Elastic-Net (bottom) problems on different data sets. The $y$-axis represents the gap between the objective value and the minimum, and the $x$-axis corresponds to running time.

the two problems, we propose the asynchronous sparse variant of AVR-SExtraGD by bringing our algorithm into a sparse asynchronous framework and compare its performance with KroMagnon (Mania et al. (2015)) and ASAGA (Leblond et al. (2016)) on rcv1 and real-sim, as shown in Table 2. The results are shown in Figure 2, which verify that the asynchronous variant of AVR-SExtraGD significantly outperforms the variants of SVRG (i.e., KroMagnon) and SAGA (Defazio et al. (2014)) (i.e., ASAGA) in terms of iterations and running time. Then, for a more comprehensive comparison, we compare the performance of more algorithms for Lasso and the $\ell_1$-norm regularized logistic regression on a9a and Covtype, and the results are shown as Figure 4 and Figure 5 in Appendix E.

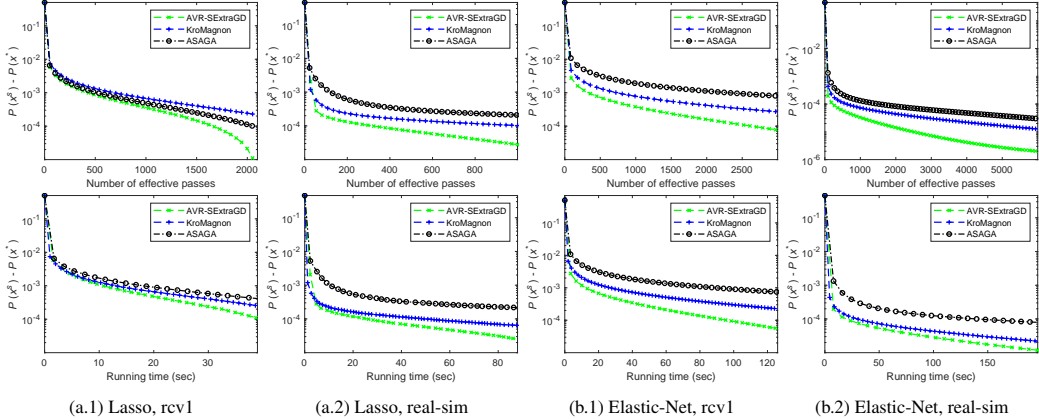

(a.1) Lasso, rcv1  (a.2) Lasso, real-sim  (b.1) Elastic-Net, rcv1  (b.2) Elastic-Net, real-sim

Figure 2: Comparison of experimental results of different algorithms for Lasso (the first two columns) and Elastic-Net (the latter two columns) problems on different data sets. The $y$-axis represents the gap of objective value, and the $x$-axis corresponds to the number of effective passes (top) or running time (bottom).

## 5.2 RESULTS ON FACE RECOGNITION

We also apply our AVR-SExtraGD as well as Prox-SVRG and Katyusha to robust face recognition via sparse representation (John et al. (2009)) on the AR and Yale. We set the loss function

in the training process as the same function as the Lasso and Elastic-Net problems. For approximately equal time, the number of outer loops is 200 for Prox-SVRG and AVR-SExtraGD, and 50 for Katyusha. In order to compare the results reasonably, we implement all the algorithms for 20 times and get the average and standard deviation of recognition rates, as shown in Table 1.

Table 1: Comparison of Recognition Rates on the AR and Yale Datasets.

| Problems | Algorithms | AR | Yale |
|---|---|---|---|
| Lasso | Prox-SVRG | $0.6000 \pm 0.0648$ | $0.6200 \pm 0.0447$ |
| | Katyusha | $0.5880 \pm 0.0832$ | $0.7480 \pm 0.0415$ |
| | AVR-SExtraGD | $0.6200 \pm 0.0756$ | $0.8100 \pm 0.0265$ |
| Elastic-Net | Prox-SVRG | $0.5630 \pm 0.0580$ | $0.6400 \pm 0.0394$ |
| | Katyusha | $0.5560 \pm 0.0738$ | $0.6870 \pm 0.0433$ |
| | AVR-SExtraGD | $0.5800 \pm 0.0634$ | $0.7140 \pm 0.0297$ |

The results in Table 1 show that the recognition rate of AVR-SExtraGD is significantly higher than other algorithms on both the AR and Yale data sets. This means that our AVR-SExtraGD can learn a more efficient representation for face recognition.

### 5.3 THE SELECTION OF $K$ IN AVR-SEXTRAGD

For the selection of $K$, we do some relevant experiments as examples. We choose different $K$ to solve Lasso problem by our algorithm, and get the results as shown in Figure 3.

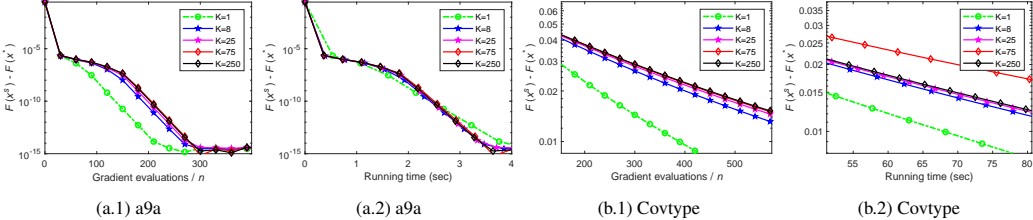

(a.1) a9a      (a.2) a9a      (b.1) Covtype      (b.2) Covtype

Figure 3: Comparison of experimental results about different choices of $K$ in AVR-SExtraGD for Lasso on different data sets. The $y$-axis represents the gap of objective value, and the $x$-axis corresponds to the number of effective passes ((a.1) and (b.1)) or running time ((a.2) and (b.2)).

where, $K = n$ means that the extragradient is calculated every integer multiple of $n$ ($n \in \{1, 8, 25, 75, 250\}$). For a9a, when the extragradient is used every time, the function value decreases faster with respect to the number of iterations, but the result is not good for running time. Thus, for a9a, we choose $K = 25$. As for Covtype, obviously, $K = 1$ is the best choice.

## 6 CONCLUSIONS AND FUTURE WORK

In this paper, we mainly considered the non-smooth optimization problem in large-scale and high-dimensional settings. By introducing the idea of extragradient and momentum acceleration, we improved the classical Prox-SVRG and then proposed a novel algorithm, called AVR-SExtraGD. From our theoretical analysis, we can know that AVR-SExtraGD attains linear convergence for SC problems, and achieves the same oracle complexity as Katyusha, which is the best-known one of stochastic first-order algorithms in both SC and non-SC cases. Finally, the experimental results showed that AVR-SExtraGD improved the result of the gap of the optimal value introduced by proximal operator, and thus improved the accuracy of solutions and convergence speed, which confirmed the efficiency of extragradient and momentum acceleration. For future work, we can extend the ideas introduced in this paper to many existing proximal algorithms, including Prox-AFG (Beck & Teboulle (2009)), Prox-SAG (Schmidt et al. (2017)) and Prox-SDCA (Shalev-Shwartz & Tong (2012); Shalev-shwartz & Tong (2014)) which is a proximal variant of SDCA (Shalev-Shwartz & Tong (2013)), and it will certainly improve the performance of these algorithms. Moreover, we can also rewrite our algorithm into the form of mini-batch, whose computation of gradient evaluations can be parallelized (Agarwal & Duchi (2011); Dekel et al. (2012)).

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

APPENDIX

## A  THE ERROR OF OPTIMAL VALUE

In this part, we prove that Prox-FG will cause the deviation of the optimal value. Based on its update rules, we can explain why Problem (1) can be solved by proximal operators and find out the reason for the introduction of the error. According to Prox-FG, we have

$$
\begin{aligned}
x_k &= \text{Prox}_{\eta_k}^R (x_{k-1} - \eta_k \nabla F(x_{k-1})) \\
&= \underset{u \in \mathbb{R}^d}{\arg\min} \left\{ R(u) + \frac{1}{2\eta_k} \| u - (x_{k-1} - \eta_k \nabla F(x_{k-1})) \|^2 \right\} \\
&= \underset{u \in \mathbb{R}^d}{\arg\min} \left\{ R(u) + \frac{\eta_k}{2} \|\nabla F(x_{k-1})\|^2 + \nabla F(x_{k-1})^T (u - x_{k-1}) + \frac{1}{2\eta_k} \| u - x_{k-1} \|^2 \right\} \\
&= \underset{u \in \mathbb{R}^d}{\arg\min} \left\{ R(u) + F(x_{k-1}) + \nabla F(x_{k-1})^T (u - x_{k-1}) + \frac{1}{2\eta_k} \| u - x_{k-1} \|^2 \right\} \\
&\approx \underset{u \in \mathbb{R}^d}{\arg\min} \left\{ F(u) + R(u) \right\}.
\end{aligned}
$$

The final approximation is obtained by the second-order Taylor expansion of $F(u)$ at $x_{k-1}$. From the above analysis, we can see that when using proximal operators to solve the problems with the $\ell_1$-norm regularization, the iterate $x_k$ in each iteration is an estimation of the optimal point, not the real optimal point. Therefore, in the last iteration, the final output point is also an estimation of $P_*$, which results in the deviation of the optimal value.

## B  PROOFS OF AVR-SEXTRAGD

### B.1  KEY LEMMAS

**Lemma 1.** *If two vectors $x_k$, $x_{k-1} \in \mathbb{R}^d$ satisfy the following equality,*

$$
x_k = \arg\min_x \left\{ \frac{1}{2\eta} \| x - x_{k-1} \|^2 + \langle \tilde{\nabla}_{i_k} F(x_{k-1}), x \rangle + R(x) \right\}
$$

*with a constant vector $\tilde{\nabla}_{i_k} F(x_{k-1})$ and a convex function $R(\cdot)$, then for $\forall u \in \mathbb{R}^d$, we have*

$$
\langle \tilde{\nabla}_{i_k} F(x_{k-1}), x_k - u \rangle \leq -\frac{1}{2\eta} \| x_{k-1} - x_k \|^2 + \frac{1}{2\eta} \| x_{k-1} - u \|^2 - \frac{1}{2\eta} \| x_k - u \|^2 + R(u) - R(x_k).
$$

*Moreover, if $R(\cdot)$ is $\mu$-strongly convex, the above inequality becomes*

$$
\langle \tilde{\nabla}_{i_k} F(x_{k-1}), x_k - u \rangle \leq -\frac{1}{2\eta} \| x_{k-1} - x_k \|^2 + \frac{1}{2\eta} \| x_{k-1} - u \|^2 - \frac{1 + \eta\mu}{2\eta} \| x_k - u \|^2 + R(u) - R(x_k).
$$

**Lemma 2** (Variance Bound). *Suppose each function $f_i(\cdot)$ is $L_i$-smooth, let $\tilde{\nabla}_{i_k} F(x_{k-1}) = \nabla_{i_k} F(x_{k-1}) - \nabla_{i_k} F(\tilde{x}^{s-1}) + \nabla F(\tilde{x}^{s-1})$, which is the gradient estimation operator used in Algorithm 1. Then the following inequality holds*

$$
\mathbb{E} \| \nabla F(x_{k-1}) - \tilde{\nabla}_{i_k} F(x_{k-1}) \|^2 \leq 2L (F(\tilde{x}^{s-1}) - F(x_{k-1}) - \langle \nabla F(x_{k-1}), \tilde{x}^{s-1} - x_{k-1} \rangle).
$$

The detailed proof of Lemma 1 can be found in Kaiwen et al. (2018), and the proof of Lemma 2 can be seen in Zeyuan (2017), and thus we omit the proofs here. Next, we give the proofs of Theorems 1 and 2.

### B.2 PROOF OF THEOREM 1

*Proof.* In this part, we consider one particular epoch and omit the number of outer iteration $s$ (except $\tilde{x}^{s-1}$ and $\tilde{x}^s$). We assume the parameters $\eta$ and $\beta$ satisfy the following inequality,

$$L\beta + \frac{L\beta}{1-\beta} \leq \frac{1}{\eta}. \tag{4}$$

Let $\hat{x}_{k-1} = \beta x_{k-1} + (1-\beta)\tilde{x}^{s-1}$, $\hat{x}_{k-1/2} = \beta x_{k-1/2} + (1-\beta)\tilde{x}^{s-1}$. Thus, we can obtain $\hat{x}_{k-1/2} - \hat{x}_{k-1} = \beta(x_{k-1/2} - x_{k-1})$. Thus, according to the $L$-smoothness of $F(\cdot)$, we can obtain

$$P(\hat{x}_{k-1/2}) \leq \beta R(x_{k-1/2}) + (1-\beta)R(\tilde{x}^{s-1}) + F(\hat{x}_{k-1}) + \langle \tilde{\nabla}_{i_k} F(\hat{x}_{k-1}), \beta(x_{k-1/2} - x_{k-1}) \rangle$$

$$+ \frac{L\beta^2}{2}\|x_{k-1/2} - x_{k-1}\|^2 + \langle \nabla F(\hat{x}_{k-1}) - \tilde{\nabla}_{i_k} F(\hat{x}_{k-1}), \beta(x_{k-1/2} - x_{k-1}) \rangle.$$

Then by using (4), we have

$$\frac{1}{\beta}P(\hat{x}_{k-1/2}) \leq R(x_{k-1/2}) + \frac{1-\beta}{\beta}R(\tilde{x}^{s-1}) + \frac{1}{\beta}F(\hat{x}_{k-1}) + \langle \tilde{\nabla}_{i_k} F(\hat{x}_{k-1}), x_{k-1/2} - x_{k-1} \rangle$$

$$+ \frac{1}{2\eta}\|x_{k-1/2} - x_{k-1}\|^2 - \frac{L\beta}{2(1-\beta)}\|x_{k-1/2} - x_{k-1}\|^2$$

$$+ \langle \nabla F(\hat{x}_{k-1}) - \tilde{\nabla}_{i_k} F(\hat{x}_{k-1}), x_{k-1/2} - x_{k-1} \rangle.$$

According to Lemma 1 with $x_{k-1}$, $x_k = x_{k-1/2}$, $u = x_*$ and using the Young's inequality to expand $\langle \nabla F(\hat{x}_{k-1}) - \tilde{\nabla}_{i_k} F(\hat{x}_{k-1}), x_{k-1/2} - x_{k-1} \rangle$ with the parameter $\theta > 0$ and taking expectation with respect to the sample $i_k$, we have

$$\frac{1}{\beta}\mathbb{E}P(\hat{x}_{k-1/2}) \leq R(x_*) + \frac{1-\beta}{\beta}R(\tilde{x}^{s-1}) + \frac{1}{\beta}F(\hat{x}_{k-1}) + \mathbb{E}\langle \tilde{\nabla}_{i_k} F(\hat{x}_{k-1}), x_* - x_{k-1} \rangle$$

$$+ \frac{1}{2\eta}\|x_* - x_{k-1}\|^2 - \frac{1+\eta\mu}{2\eta}\mathbb{E}\|x_* - x_{k-1/2}\|^2 - \frac{L\beta}{2(1-\beta)}\mathbb{E}\|x_{k-1/2} - x_{k-1}\|^2$$

$$+ \frac{\theta}{2}\mathbb{E}\|x_{k-1/2} - x_{k-1}\|^2 + \frac{1}{2\theta}\mathbb{E}\|\nabla F(\hat{x}_{k-1}) - \tilde{\nabla}_{i_k} F(\hat{x}_{k-1})\|^2.$$

We set $\theta = \frac{L\beta}{1-\beta} > 0$ and apply Lemma 2, then

$$\frac{1}{\beta}\mathbb{E}P(\hat{x}_{k-1/2}) \leq R(x_*) + \frac{1-\beta}{\beta}R(\tilde{x}^{s-1}) + \frac{1}{\beta}F(\hat{x}_{k-1})$$

$$+ \frac{1}{2\eta}\|x_* - x_{k-1}\|^2 - \frac{1+\eta\mu}{2\eta}\mathbb{E}\|x_* - x_{k-1/2}\|^2 + \frac{1-\beta}{\beta}[F(\tilde{x}^{s-1}) - F(\hat{x}_{k-1})]$$

$$+ \mathbb{E}\langle \tilde{\nabla}_{i_k}F(\hat{x}_{k-1}), x_* + \frac{1-\beta}{\beta}\tilde{x}^{s-1} - \frac{1}{\beta}\hat{x}_{k-1} + \frac{1-\beta}{\beta}(\hat{x}_{k-1} - \tilde{x}^{s-1})\rangle$$

$$\leq R(x_*) + \frac{1-\beta}{\beta}R(\tilde{x}^{s-1}) + \frac{1}{\beta}F(\hat{x}_{k-1})$$

$$+ \frac{1}{2\eta}\|x_* - x_{k-1}\|^2 - \frac{1+\eta\mu}{2\eta}\mathbb{E}\|x_* - x_{k-1/2}\|^2 + \frac{1-\beta}{\beta}[F(\tilde{x}^{s-1}) - F(\hat{x}_{k-1})]$$

$$+ \frac{1}{\beta}\mathbb{E}\langle \tilde{\nabla}_{i_k}F(\hat{x}_{k-1}), \beta x_* + (1-\beta)\hat{x}_{k-1} - \hat{x}_{k-1}\rangle$$

$$\leq R(x_*) + \frac{1-\beta}{\beta}R(\tilde{x}^{s-1}) + \frac{1}{\beta}F(\hat{x}_{k-1})$$

$$+ \frac{1}{2\eta}\|x_* - x_{k-1}\|^2 - \frac{1+\eta\mu}{2\eta}\mathbb{E}\|x_* - x_{k-1/2}\|^2 + \frac{1-\beta}{\beta}[F(\tilde{x}^{s-1}) - F(\hat{x}_{k-1})]$$

$$+ \frac{1}{\beta}F(\beta x_* + (1-\beta)\hat{x}_{k-1}) - \frac{1}{\beta}F(\hat{x}_{k-1})$$

$$\leq R(x_*) + \frac{1-\beta}{\beta}R(\tilde{x}^{s-1}) + \frac{1}{\beta}F(\hat{x}_{k-1}) + \frac{1-\beta}{\beta}[F(\tilde{x}^{s-1}) - F(\hat{x}_{k-1})] + F(x_*)$$

$$+ \frac{1-\beta}{\beta}F(\hat{x}_{k-1}) - \frac{1}{\beta}F(\hat{x}_{k-1}) + \frac{1}{2\eta}\|x_* - x_{k-1}\|^2 - \frac{1+\eta\mu}{2\eta}\mathbb{E}\|x_* - x_{k-1/2}\|^2$$

$$= \frac{1-\beta}{\beta}P(\tilde{x}^{s-1}) + P(x_*) + \frac{1}{2\eta}\|x_* - x_{k-1}\|^2 - \frac{1+\eta\mu}{2\eta}\mathbb{E}\|x_* - x_{k-1/2}\|^2.$$

The third inequality holds due to $\mathbb{E}[\tilde{\nabla}_{i_k}F(\hat{x}_{k-1})] = \nabla F(\hat{x}_{k-1})$ and the convexity of $F(\cdot)$, then we get

$$\frac{1}{\beta}\mathbb{E}[P(\hat{x}_{k-1/2}) - P(x_*)] \leq \frac{1-\beta}{\beta}[P(\tilde{x}^{s-1}) - P(x_*)] + \frac{1}{2\eta}\|x_* - x_{k-1}\|^2 - \frac{1+\eta\mu}{2\eta}\mathbb{E}\|x_* - x_{k-1/2}\|^2.$$

Moreover, because $\hat{x}_{k-1} = \beta x_{k-1} + (1-\beta)\tilde{x}^{s-1}$, we can obtain $\hat{x}_k = \beta x_k + (1-\beta)\tilde{x}^{s-1}$. Thus, it is not hard to know

$$\frac{1}{\beta}\mathbb{E}[P(\hat{x}_k) - P(x_*)] \leq \frac{1-\beta}{\beta}[P(\tilde{x}^{s-1}) - P(x_*)] + \frac{1}{2\eta}\|x_* - x_{k-1/2}\|^2 - \frac{1+\eta\mu}{2\eta}\mathbb{E}\|x_* - x_k\|^2.$$

We set $y_k = \beta\frac{x_{k-1/2} + x_k}{2} + (1-\beta)\tilde{x}^{s-1}$ and it is obvious that $\frac{1}{2\eta} - \frac{1+\eta\mu}{2\eta} \leq 0$, then we have

$$\frac{1}{\beta}\mathbb{E}[P(y_k) - P(x_*)] \leq \frac{1-\beta}{\beta}[P(\tilde{x}^{s-1}) - P(x_*)] + \frac{1}{4\eta}\|x_* - x_{k-1}\|^2 - \frac{1+\eta\mu}{4\eta}\mathbb{E}\|x_* - x_k\|^2. \quad (5)$$

By setting $\rho = 1 + \eta\mu$ and summing (5) over $k = 1, ..., m$ with increasing weight $\rho^{k-1}$, we have

$$\frac{1}{\beta}\sum_{k=1}^{m}\rho^{k-1}\mathbb{E}[P(y_k)-P(x_*)]+\frac{\rho^m}{4\eta}\|x_m-x_*\|^2 \leq \frac{1-\beta}{\beta}\sum_{k=1}^{m}\rho^{k-1}[P(\tilde{x}^{s-1})-P(x_*)]+\frac{1}{4\eta}\|x_0-x_*\|^2.$$

Because, for SC problems, we set $\tilde{x}^s = (\sum_{k=1}^{m}\rho_{k-1})^{-1}\sum_{k=1}^{m}\rho^{k-1}y_k$ in Algorithm 1 , we have

$$\frac{1}{\beta}\sum_{k=1}^{m}\rho^{k-1}\mathbb{E}[P(\tilde{x}^s)-P(x_*)]+\frac{\rho^m}{4\eta}\|x_m^s-x_*\|^2 \leq \frac{1-\beta}{\beta}\sum_{k=1}^{m}\rho^{k-1}[P(\tilde{x}^{s-1})-P(x_*)]+\frac{1}{4\eta}\|x_0^s-x_*\|^2.$$

Then, according to the convergence analysis for SC problems in (Kaiwen et al. (2018)), we can get a similar result. That is, for the case with $\frac{m}{\kappa} \leq \frac{3}{4}$, we set $\eta = \sqrt{\frac{1}{3\mu mL}}, \beta = \sqrt{\frac{m}{3\kappa}} \leq \frac{1}{2}$, and $m = \Theta(n)$, then we can obtain

$$\mathbb{E}[P(\tilde{x}^S)-P(x_*)] \leq \left(O(1+\sqrt{\frac{1}{3n\kappa}})\right)^{-Sm}\cdot O(P(\tilde{x}^0)-P(x_*)).$$

We note that $\tilde{x}^0 = x_0$, so we get

$$\mathbb{E}[P(\tilde{x}^S)-P(x_*)] \leq \left(O(1+\sqrt{\frac{1}{3n\kappa}})\right)^{-Sm}\cdot O(P(x_0)-P(x_*)),$$

which implies that the oracle complexity in this case to achieve an $\epsilon$-additive error is $\mathcal{O}(\sqrt{\kappa n}\log\frac{P(x_0)-P(x_*)}{\epsilon})$. However, for the case with $\frac{m}{\kappa} > \frac{3}{4}$, we set $\eta = \frac{2}{3L}, \beta = \frac{1}{2}$ and $m = \Theta(n)$, then we can obtain

$$\mathbb{E}[P(\tilde{x}^S)-P(x_*)] \leq \left(\frac{2}{3}\right)^{S}\cdot O(P(\tilde{x}^0)-P(x_*)).$$

We know that $\tilde{x}^0 = x_0$, so we have

$$\mathbb{E}[P(\tilde{x}^S)-P(x_*)] \leq \left(\frac{2}{3}\right)^{S}\cdot O(P(x_0)-P(x_*)),$$

which implies that the oracle complexity of AVR-SExtraGD in this case is $\mathcal{O}\left(n\log\frac{P(x_0)-P(x_*)}{\epsilon}\right)$.

$\square$

### B.3 PROOF OF THEOREM 2

*Proof.* In this part, we also omit the number of outer iteration $s$ (except $\tilde{x}^{s-1}$ and $\tilde{x}^s$). Due to $\hat{x}_{k-1/2} = \beta x_{k-1/2} + (1-\beta)\tilde{x}^{s-1}$, we can get

$$P(\hat{x}_{k-1/2}) = P(\beta x_{k-1/2}+(1-\beta)\tilde{x}^{s-1}) = R(\beta x_{k-1/2}+(1-\beta)\tilde{x}^{s-1})+F(\hat{x}_{k-1/2}).$$

From the convexity of $R(\cdot)$ and $L$-smoothness of $F(\cdot)$, we obtain

$$P(\hat{x}_{k-1/2}) \leq \beta R(x_{k-1/2}) + (1-\beta)R(\tilde{x}^{s-1}) + F(\hat{x}_{k-1}) + \langle \tilde{\nabla}_{i_k} F(\hat{x}_{k-1}), \beta(x_{k-1/2} - x_{k-1}) \rangle$$

$$+ \frac{L\beta^2}{2} \|x_{k-1/2} - x_{k-1}\|^2 + \langle \nabla F(\hat{x}_{k-1}) - \tilde{\nabla}_{i_k} F(\hat{x}_{k-1}), \beta(x_{k-1/2} - x_{k-1}) \rangle$$

$$\leq \beta R(x_{k-1/2}) + (1-\beta)R(\tilde{x}^{s-1}) + F(\hat{x}_{k-1}) + \langle \tilde{\nabla}_{i_k} F(\hat{x}_{k-1}), \beta(x_{k-1/2} - x_{k-1}) \rangle$$

$$+ \frac{L\alpha\beta^2}{2} \|x_{k-1/2} - x_{k-1}\|^2 + \frac{1}{2L(\alpha-1)} \|\nabla F(\hat{x}_{k-1}) - \tilde{\nabla}_{i_k} F(\hat{x}_{k-1})\|^2,$$

where the second inequality holds by using the Young's inequality with the parameter $L(\alpha - 1)$,

where $\alpha$ is a small constant. After applying Lemma 1 and taking expectation with respect to the

sample $i_k$, we have

$$
\begin{aligned}
\mathbb{E}P(\hat{x}_{k-1/2}) \leq {} & \beta R(x_*) + (1-\beta)R(\tilde{x}^{s-1}) + F(\hat{x}_{k-1}) + \mathbb{E}\langle\tilde{\nabla}_{i_k}F(\hat{x}_{k-1}), \beta(x_* - x_{k-1})\rangle \\
& + \frac{L\alpha\beta^2}{2}(\|x_* - x_{k-1}\|^2 - \mathbb{E}\|x_* - x_{k-1/2}\|^2) \\
& + \frac{1}{2L(\alpha-1)}\mathbb{E}\|\nabla F(\hat{x}_{k-1}) - \tilde{\nabla}_{i_k}F(\hat{x}_{k-1})\|^2 \\
\leq {} & \beta R(x_*) + (1-\beta)R(\tilde{x}^{s-1}) + F(\hat{x}_{k-1}) + \mathbb{E}\langle\tilde{\nabla}_{i_k}F(\hat{x}_{k-1}), \beta(x_* - x_{k-1})\rangle \\
& + \frac{L\alpha\beta^2}{2}(\|x_* - x_{k-1}\|^2 - \mathbb{E}\|x_* - x_{k-1/2}\|^2) \\
& + \frac{1}{\alpha-1}[F(\tilde{x}^{s-1}) - F(\hat{x}_{k-1}) + \mathbb{E}\langle\tilde{\nabla}_{i_k}F(\hat{x}_{k-1}), \hat{x}_{k-1} - \tilde{x}^{s-1}\rangle] \\
\leq {} & \beta R(x_*) + (1-\beta)R(\tilde{x}^{s-1}) + F(\hat{x}_{k-1}) + \frac{1}{\alpha-1}[F(\tilde{x}^{s-1}) - F(\hat{x}_{k-1})] \\
& + \mathbb{E}\langle\tilde{\nabla}_{i_k}F(\hat{x}_{k-1}), \beta x_* + (1-\beta)\tilde{x}^{s-1} - \hat{x}_{k-1} + \frac{1}{\alpha-1}(\hat{x}_{k-1} - \tilde{x}^{s-1})\rangle \\
& + \frac{L\alpha\beta^2}{2}(\|x_* - x_{k-1}\|^2 - \mathbb{E}\|x_* - x_{k-1/2}\|^2) \\
\leq {} & \beta R(x_*) + (1-\beta)R(\tilde{x}^{s-1}) + F(\hat{x}_{k-1}) + \frac{1}{\alpha-1}[F(\tilde{x}^{s-1}) - F(\hat{x}_{k-1})] \\
& + F(\beta x_* + (1-\beta-\frac{1}{\alpha-1})\tilde{x}^{s-1} + \frac{1}{\alpha-1}\hat{x}_{k-1}) - F(\hat{x}_{k-1}) \\
& + \frac{L\alpha\beta^2}{2}(\|x_* - x_{k-1}\|^2 - \mathbb{E}\|x_* - x_{k-1/2}\|^2) \\
\leq {} & \beta R(x_*) + (1-\beta)R(\tilde{x}^{s-1}) + F(\hat{x}_{k-1}) + \frac{1}{\alpha-1}[F(\tilde{x}^{s-1}) - F(\hat{x}_{k-1})] \\
& + \beta F(x_*) + (1-\beta-\frac{1}{\alpha-1})F(\tilde{x}^{s-1}) + \frac{1}{\alpha-1}F(\hat{x}_{k-1}) - F(\hat{x}_{k-1}) \\
& + \frac{L\alpha\beta^2}{2}(\|x_* - x_{k-1}\|^2 - \mathbb{E}\|x_* - x_{k-1/2}\|^2) \\
= {} & (1-\beta)P(\tilde{x}^{s-1}) - \beta P(x_*) + \frac{L\alpha\beta^2}{2}(\|x_* - x_{k-1}\|^2 - \mathbb{E}\|x_* - x_{k-1/2}\|^2).
\end{aligned}
$$

The second inequality holds due to Lemma 2. The reasons why the fourth inequality holds are $\mathbb{E}[\tilde{\nabla}_{i_k}F(\hat{x}_{k-1})] = \nabla F(\hat{x}_{k-1})$ and the convexity of $F(\cdot)$. Besides, we need to assume $1-\beta-\frac{1}{\alpha-1}\geq 0$ in this step. Then, we get

$$
\mathbb{E}[P(\hat{x}_{k-1/2}) - P(x_*)] \leq (1-\beta)[P(\tilde{x}^{s-1}) - P(x_*)] + \frac{L\alpha\beta^2}{2}(\|x_* - x_{k-1}\|^2 - \mathbb{E}\|x_* - x_{k-1/2}\|^2).
$$

Moreover, because $\hat{x}_{k-1} = \beta x_{k-1} + (1-\beta)\tilde{x}^{s-1}$, we can obtain $\hat{x}_k = \beta x_k + (1-\beta)\tilde{x}^{s-1}$. Thus, it is not hard to know

$$\mathbb{E}[P(\hat{x}_k) - P(x_*)] \leq (1-\beta)[P(\tilde{x}^{s-1}) - P(x_*)] + \frac{L\alpha\beta^2}{2}(\|x_* - x_{k-1/2}\|^2 - \mathbb{E}\|x_* - x_k\|^2).$$

Let $y_k = \beta \frac{x_{k-1/2} + x_k}{2} + (1-\beta)\tilde{x}^{s-1}$, we have

$$\mathbb{E}[P(y_k) - P(x_*)] \leq (1-\beta)[P(\tilde{x}^{s-1}) - P(x_*)] + \frac{L\alpha\beta^2}{4}(\|x_* - x_{k-1}\|^2 - \mathbb{E}\|x_* - x_k\|^2).$$

That is,

$$\frac{1}{\beta^2}\mathbb{E}[P(y_k) - P(x_*)] \leq \frac{(1-\beta)}{\beta^2}[P(\tilde{x}^{s-1}) - P(x_*)] + \frac{L\alpha}{4}(\|x_* - x_{k-1}\|^2 - \mathbb{E}\|x_* - x_k\|^2).$$

Since we have $\tilde{x}^s = \frac{\beta}{m}\sum_{k=1}^m \frac{x_{k-1/2} + x_k}{2} + (1-\beta)\tilde{x}^{s-1} = \frac{1}{m}\sum_{k=1}^m y_k$ in Algorithm 1, and then by summing the previous inequality over $k = 1, ..., m$ and according to $x_0^{s+1} = x_m^s$, we obtain

$$\frac{1}{\beta_s^2}\mathbb{E}[P(\tilde{x}^s) - P(x_*)] \leq \frac{(1-\beta_s)}{\beta_s^2}[P(\tilde{x}^{s-1}) - P(x_*)] + \frac{L\alpha}{4m}(\|x_* - x_0^s\|^2 - \|x_* - x_0^{s+1}\|^2).$$

We set $\beta_s = \frac{2}{s+4}$ and can easily obtain $\frac{1}{\beta_{s-1}^2} \geq \frac{1-\beta_s}{\beta_s^2}$. Then by summing the previous inequality over $s = 1, ..., S$, we can get

$$\frac{1}{\beta_S^2}\mathbb{E}[P(\tilde{x}^S) - P(x_*)] \leq \frac{(1-\beta_1)}{\beta_1^2}[P(\tilde{x}^{s-1}) - P(x_*)] + \frac{L\alpha}{4m}(\|x_* - x_0^1\|^2 - \|x_* - x_m^S\|^2).$$

Then we have

$$\mathbb{E}[P(\tilde{x}^S) - P(x_*)] \leq \frac{4(1-\beta_1)}{(S+4)^2\beta_1^2}(P(\tilde{x}_0) - P(x_*)) + \frac{L\alpha}{(S+4)^2 m}\|\tilde{x}_0 - x_*\|^2,$$

which holds because $\|x_* - x_m^S\|^2 \geq 0$. We note that $\tilde{x}^0 = x_0^1 = x_0$, so we get

$$\mathbb{E}[P(\tilde{x}^S) - P(x_*)] \leq \frac{4(1-\beta_1)}{(S+4)^2\beta_1^2}(P(x_0) - P(x_*)) + \frac{L\alpha}{(S+4)^2 m}\|x_0 - x_*\|^2.$$

In other words, by choosing $m = \Theta(n)$, the total oracle complexity is

$$\mathcal{O}\left(n\sqrt{\frac{P(x_0) - P(x_*)}{\epsilon}} + \sqrt{\frac{nL\|x_0 - x_*\|^2}{\epsilon}}\right).$$

$\square$

Finally, we finished the convergence analysis of AVR-SExtraGD. In our Algorithm 1, only part of the iterations are updated by extragradient descent, and the other iterations are updated with the update rules of MiG. And we know both MiG and AVR-SExtraGD can make the objective function converge to $P_*$. Therefore, the method in Algorithm 1 will not affect the results and can accelerate the algorithm.

---

**Algorithm 2** VR-SExtraGD

---

**Input:** Initial vector $x_0$, the number of epochs $S$, the number of iterations $m$ per epoch, and the step sizes $\eta_1, \eta_2$.

**Initialize:** $\tilde{x}^0 = x^0$.

1: **for** $s = 1, 2, \ldots, S$ **do**
2:  $\quad \tilde{\mu}^{s-1} = \nabla F(\tilde{x}^{s-1})$;
3:  $\quad x_0^s = \tilde{x}^{s-1}$ (SC) or $x_0^s = x_m^{s-1}$ (non-SC);
4:  $\quad$ **for** $k = 1, 2, \ldots, m$ **do**
5:  $\quad\quad$ Pick $i_k$ uniformly at random from $\{1, ..., n\}$;
6:  $\quad\quad \tilde{\nabla} f_{i_k}(x_{k-1}^s) = \nabla f_{i_k}(x_{k-1}^s) - \nabla f_{i_k}(\tilde{x}^{s-1}) + \tilde{\mu}^{s-1}$;
7:  $\quad\quad x_{k-1/2}^s = \mathrm{Prox}_{\eta_1}^R(x_{k-1}^s - \eta_1 \tilde{\nabla} f_{i_k}(x_{k-1}^s))$;
8:  $\quad\quad \tilde{\nabla} f_{i_k}(x_{k-1/2}^s) = \nabla f_{i_k}(x_{k-1/2}^s) - \nabla f_{i_k}(\tilde{x}^{s-1}) + \tilde{\mu}^{s-1}$;
9:  $\quad\quad x_k^s = \mathrm{Prox}_{\eta_2}^R(x_{k-1/2}^s - \eta_2 \tilde{\nabla} f_{i_k}(x_{k-1/2}^s))$;
10: $\quad$ **end for**
11: $\quad \tilde{x}^s = \frac{1}{m} \sum_{k=1}^m x_k^s$;
12: **end for**

**Output:** $\tilde{x}^S$.

---

## C  SEXTRAGD AND VR-SEXTRAGD

Based on the update rules of EEG, we first consider and propose the stochastic variant of the algorithm, namely the stochastic extragradient descent (SExtraGD) algorithm, and its variance reduced variant called *variance reduced stochastic extragradient descent* (VR-SExtraGD), whose update rules can be formulated as follows:

- The update rules of SExtraGD:
$$x_{k-1/2} = \mathrm{Prox}_{\eta_1}^R(x_{k-1} - \eta_1 \nabla f_{i_k}(x_{k-1})); \quad x_k = \mathrm{Prox}_{\eta_2}^R\left(x_{k-1/2} - \eta_2 \nabla f_{i_k}(x_{k-1/2})\right).$$

- The update rules of VR-SExtraGD:
$$x_{k-1/2} = \mathrm{Prox}_{\eta_1}^R\left(x_{k-1} - \eta_1 \tilde{\nabla} f_{i_k}(x_{k-1})\right); \quad x_k = \mathrm{Prox}_{\eta_2}^R\left(x_{k-1/2} - \eta_2 \tilde{\nabla} f_{i_k}(x_{k-1/2})\right),$$

where $\tilde{\nabla} f_{i_k}(\cdot)$ is the gradient estimation defined in (3).

We note that one difference between these two algorithms and EEG is that we change $x_{k-1}$ in Step two to $x_{k-1/2}$, which will be beneficial to our theoretical analysis, but will not cause any major change to the results of the algorithms. Thus, we propose the stochastic extragradient descent algorithms, called SExtraGD and VR-SExtraGD, for solving non-smooth (both SC and non-SC) problems. In addition, the detailed process of VR-SExtraGD is shown as outlined in **Algorithm 2**.

## D  CONVERGENCE ANALYSIS OF VR-SEXTRAGD

Firstly, we give some key lemmas which are helpful for the convergence analysis of VR-SExtraGD. Lemmas 3, 4 and 5 are used to prove Lemma 6 which is an important lemma to prove the convergence of VR-SExtraGD.

**Lemma 3.** *Let $R(\cdot)$ be a convex function from $\mathbb{R}^d$ to $\mathbb{R}$, and $\eta_k > 0$. Then, for all $x, y \in \mathbb{R}^d$,*

$$\|\mathrm{Prox}_{\eta_k}^R(x) - \mathrm{Prox}_{\eta_k}^R(y)\| \leq \|x - y\|.$$

**Lemma 4.** *Let $P(x) = F(x) + R(x)$, and $\nabla F(x)$ is Lipschitz continuous with parameter L. For any $x \in \mathbb{R}^d$ and arbitrary $v \in \mathbb{R}^d$, we define*

$$x' = \text{Prox}_\eta^R(x - \eta v), \quad g = \frac{1}{\eta}(x - x'), \quad \triangle = v - \nabla F(x),$$

*where $\eta$ is a step size that satisfies $0 < \eta \leq \frac{1}{L}$. Thus, we can know that for any $y \in \mathbb{R}^d$,*

$$P(y) \geq P(x') + g^T(y - x) + \frac{\eta}{2}\|g\|^2 + \triangle^T(x' - y).$$

**Lemma 5.** *Considering $P(x)$ as defined in Problem (1) and $\tilde{\nabla} f_{i_k}(v)$ as defined in (3), where $v$ is an arbitrary stochastic sample, and let $x_* = \arg\min_x P(x)$. We have $\mathbb{E}[\tilde{\nabla} f_{i_k}(v)] = \nabla F(v)$ and*

$$\mathbb{E}\|\tilde{\nabla} f_{i_k}(v) - \nabla F(v)\|^2 \leq 4L[P(v) - P(x_*) + P(\tilde{x}) - P(x_*)].$$

Because Lemma 3 is well known and often used (e.g., see Section 3 in Rockafellar (1970)), we omit the proof of this lemma here. For Lemma 4, it is very similar to Lemma 3.7 of Lin & Tong (2014), and thus can be easily proved, so we also omit the proof here. Similarly, according to Lemma 3.4 in Lin & Tong (2014), Lemma 5 can be also easily proved, and thus we will not give the detail about it. Then we can prove the following lemma by these three lemmas.

**Lemma 6.** *For all randomly selected sample $v$ and $h$, if*

$$v = \text{Prox}_\eta^R\left(h - \eta\tilde{\nabla} f_{i_k}(h)\right), \tag{6}$$

*we have*

$$\mathbb{E}\|v - x_*\|^2 \leq \|h - x_*\|^2 - 2\eta[\mathbb{E}P(v) - P(x_*)] + 8L\eta^2[P(h) - P(x_*) + P(\tilde{x}) - P(x_*)],$$

*where $x_* = \arg\min_x P(x)$.*

*Proof.* First, we define a stochastic gradient mapping as follows:

$$g = \frac{1}{\eta}(h - v) = \frac{1}{\eta}\left(h - \text{Prox}_\eta^R\left(h - \eta\tilde{\nabla} f_{i_k}(h)\right)\right).$$

According to this definition, (6) can be expressed more succinctly as follows:

$$v = h - \eta g.$$

Then we consider the distance between $v$ and $x_*$.

$$\|v - x_*\|^2 = \|h - \eta g - x_*\|^2 = \|h - x_*\|^2 - 2\eta g^T(h - x_*) + \eta^2\|g\|^2.$$

Applying Lemma 4 with $x = h$, $v = \tilde{\nabla} f_{i_k}(h)$, $x^+ = v$ and $y = x_*$, we have

$$-g^T(h - x_*) + \frac{\eta}{2}\|g\|^2 \leq P(x_*) - P(v) - \triangle_k^T(v - x_*)$$

where $\triangle_k = \tilde{\nabla} f_{i_k}(h) - \nabla F(h)$. Thus we have

$$\|v - x_*\|^2 \leq \|h - x_*\|^2 - 2\eta[P(v) - P(x_*)] - 2\eta\triangle_k^T(v - x_*).$$

Then we can give an upper bound of $-2\eta\triangle_k^T(v - x_*)$. First of all, we can define the update of

Prox-FG as shown below (although it is not used in our algorithm):

$$\bar{h} = \text{Prox}_\eta^R(h - \eta\nabla F(h)).$$

So, we can obtain

$$-2\eta\triangle_k^T(v - x_*) = -2\eta\triangle_k^T(v - \bar{h}) - 2\eta\triangle_k^T(\bar{h} - x_*)$$

$$\leq 2\eta\|\triangle_k\|\|v - \bar{h}\| - 2\eta\triangle_k^T(\bar{h} - x_*)$$

$$\leq 2\eta\|\triangle_k\|\|(h - \eta\tilde{\nabla} f_{i_k}(h)) - (h - \eta\nabla F(h))\| - 2\eta\triangle_k^T(\bar{h} - x_*)$$

$$= 2\eta^2\|\triangle_k\|^2 - 2\eta\triangle_k^T(\bar{h} - x_*).$$

The first inequality follows from the Cauchy-Schwarz inequality, and the second inequality holds

due to Lemma 3. So, we have

$$\|h - x_*\|^2 \leq \|v - x_*\|^2 - 2\eta[P(h) - P(x_*)] + 2\eta^2\|\triangle_k\|^2 - 2\eta\triangle_k^T(\bar{h} - x_*).$$

Then, we take expectation on both sides of the above inequality with respect to $i_k$ to obtain

$$\mathbb{E}\|h - x_*\| \leq \|v - x_*\|^2 - 2\eta[\mathbb{E}P(h) - P(x_*)] + 2\eta^2 E\|\triangle_k\|^2 - 2\eta\mathbb{E}[\triangle_k^T(\bar{h} - x_*)]. \quad (7)$$

It can be noted that both $\bar{h}$ and $x_*$ are independent of the random variable $i_k$, and we can easily

know that $\mathbb{E}\triangle_k = 0$. So

$$\mathbb{E}[\triangle_k^T(\bar{h} - x_*)] = (\mathbb{E}\triangle_k)^T(\bar{h} - x_*) = 0. \quad (8)$$

Substituting Lemma 5 and (8) into (7), we obtain

$$\mathbb{E}\|h - x_*\|^2 \leq \|v - x_*\|^2 - 2\eta[\mathbb{E}P(h) - P(x_*)] + 8L\eta^2[P(v) - P(x_*) + P(\tilde{x}) - P(x_*)]. \quad (9)$$

$$\square$$

### D.1 FOR SC PROBLEMS

Firstly, we can prove the convergence of VR-SExtraGD for strongly convex (SC) problems, which is showed by the following theorem.

**Theorem 3** (Strongly Convex). *Suppose that Assumptions 1, 2 and 3 hold, and let $x_* = \arg\min_x P(x)$. In addition, assume $\eta_1 > 0, \eta_2 > 0, \eta_1 \geq 4L\eta_2^2$ and $\eta_2 \geq 4L\eta_1^2$ and $m$ is sufficiently large so that*

$$\theta = \frac{1}{\mu m(\eta_2 - 4L\eta_1^2)} + \frac{4L[(m+1)\eta_1^2 + m\eta_2^2]}{m(\eta_2 - 4L\eta_1^2)} \leq 1.$$

*Then VR-SExtraGD outlined in Algorithm 2 achieves linear convergence in the expected form, which can be formulated as follows:*

$$\mathbb{E}[P(\tilde{x}^S) - P(x_*)] \leq \theta^S[P(x_0) - P(x_*)]. \tag{10}$$

*Proof.* According to Lemma 6 and the update rules of VR-SExtraGD, we can easily get:

$$\mathbb{E}\|x_{k-1/2} - x_*\|^2 \leq \|x_{k-1} - x_*\|^2 - 2\eta_1[\mathbb{E}P(x_{k-1/2}) - P(x_*)]$$
$$+ 8L\eta_1^2[P(x_{k-1}) - P(x_*) + P(\tilde{x}) - P(x_*)] \tag{11}$$

and

$$\mathbb{E}\|x_k - x_*\|^2 \leq \|x_{k-1/2} - x_*\|^2 - 2\eta_2[\mathbb{E}P(x_k) - P(x_*)]$$
$$+ 8L\eta_2^2[P(x_{k-1/2}) - P(x_*) + P(\tilde{x}) - P(x_*)]. \tag{12}$$

Then, we substitute (11) into (12) to obtain

$$\mathbb{E}\|x_k - x_*\|^2 \leq \|x_{k-1} - x_*\|^2 - 2\eta_1[\mathbb{E}P(x_{k-1/2}) - P(x_*)]$$

$$+ 8L\eta_1^2[P(x_{k-1}) - P(x_*) + P(\tilde{x}) - P(x_*)] - 2\eta_2[\mathbb{E}P(x_k) - P(x_*)]$$

$$+ 8L\eta_2^2[P(x_{k-1/2}) - P(x_*) + P(\tilde{x}) - P(x_*)].$$

Suppose $\eta_1 \geq 4L\eta_2^2$, i.e., $-2\eta_1 \leq -8L\eta_2^2$, we can get

$$\mathbb{E}\|x_k - x_*\|^2 \leq \|x_{k-1} - x_*\|^2 - 8L\eta_2^2[\mathbb{E}P(x_{k-\frac{1}{2}}) - P(x_*)]$$

$$+ 8L\eta_1^2[P(x_{k-1}) - P(x_*) + P(\tilde{x}) - P(x_*)] - 2\eta_2[\mathbb{E}P(x_k) - P(x_*)]$$

$$+ 8L\eta_2^2[P(x_{k-\frac{1}{2}}) - P(x_*)] + 8L\eta_2^2[P(\tilde{x}) - P(x_*)]$$

$$= \|x_{k-1} - x_*\|^2 + 8L\eta_1^2[P(x_{k-1}) - P(x_*) + P(\tilde{x}) - P(x_*)]$$

$$- 2\eta_2[\mathbb{E}P(x_k) - P(x_*)] + 8L\eta_2^2[P(\tilde{x}) - P(x_*)].$$

By summing the previous inequality over $k = 1, ..., m$, we obtain

$$\|x_m - x_*\|^2 + 2\eta_2[\mathbb{E}P(x_m) - P(x_*)] + 2(\eta_2 - 4L\eta_1^2)\sum_{k=1}^{m-1}[\mathbb{E}P(x_k) - P(x_*)]$$

$$\leq \|x_0 - x_*\|^2 + 8L\eta_1^2[P(x_0) - P(x_*)] + m8L(\eta_1^2 + \eta_2^2)[P(\tilde{x}) - P(x_*)].$$

Since $\eta_2 - 4L\eta_1^2 < \eta_2$, and from the algorithm of VR-SExtraGD, we know $x_0 = \tilde{x}$. Therefore,

$$2(\eta_2 - 4L\eta_1^2)\sum_{k=1}^{m}[\mathbb{E}P(x_k) - P(x_*)] \leq \|\tilde{x} - x_*\|^2 + 8L[(m+1)\eta_1^2 + m\eta_2^2][P(\tilde{x}) - P(x_*)].$$

Because in a fixed epoch, such as the $s$-th epoch, there are $\tilde{x}^s = \frac{1}{m}\sum_{k=1}^{m} x_k$ and $\tilde{x}^{s-1} = x_0$, and

according to the convexity of $P(\cdot)$, $P(\tilde{x}^s) \leq \frac{1}{m}\sum_{k=1}^{m} P(x_k)$ can be obtained. Therefore,

$$2(\eta_2 - 4L\eta_1^2)m[\mathbb{E}P(\tilde{x}^s) - P(x_*)] \leq 8L[(m+1)\eta_1^2 + m\eta_2^2][P(\tilde{x}^{s-1}) - P(x_*)] + \|\tilde{x}^{s-1} - x_*\|^2. \quad (13)$$

Because of the convexity of $F(\cdot)$ and the strong convexity of $R(\cdot)$, we know $P(\cdot)$ is also strongly

convex, then we have $\|\tilde{x}^{s-1} - x_*\|^2 \leq \frac{2}{\mu}[P(\tilde{x}^{s-1}) - P(x_*)]$. Thus,

$$2(\eta_2 - 4L\eta_1^2)m[\mathbb{E}P(\tilde{x}^s) - P(x_*)] \leq (\frac{2}{\mu} + 8L((m+1)\eta_1^2 + m\eta_2^2))[P(\tilde{x}^{s-1}) - P(x_*)],$$

which is equivalent to

$$\mathbb{E}[P(\tilde{x}^s) - P(x_*)] \leq \theta[P(\tilde{x}^{s-1}) - P(x_*)],$$

where

$$\theta = \frac{1}{\mu m(\eta_2 - 4L\eta_1^2)} + \frac{4L[(m+1)\eta_1^2 + m\eta_2^2]}{m(\eta_2 - 4L\eta_1^2)}.$$

At last, we have

$$\mathbb{E}[P(\tilde{x}^S) - P(x_*)] \leq \theta\mathbb{E}[P(\tilde{x}^{S-1}) - P(x_*)]$$

$$\leq \theta^2\mathbb{E}[P(\tilde{x}^{S-2}) - P(x_*)]$$

$$\leq ... \leq \theta^S[P(\tilde{x}^0) - P(x_*)].$$

We note that $\tilde{x}^0 = x_0$, so we can obtain

$$\mathbb{E}[P(\tilde{x}^S) - P(x_*)] \leq \theta^S[P(x_0) - P(x_*)]$$

$\square$

## D.2 FOR NON-SC PROBLEMS

We can also use a theorem to give the convergence of VR-SExtraGD for solving non-SC problems, as shown below.

**Theorem 4** (Non-Strongly Convex). *Suppose that Assumptions 1 and 2 hold, and let $x_* = \arg\min_x P(x)$. In addition, assume $\eta_1 > 0, \eta_2 > 0$, and $\eta_1 = \eta_2 = \eta = \frac{1}{L\alpha}$. Then, the convergence property of VR-SExtraGD, as outlined in Algorithm 2, is given as follows:*

$$\mathbb{E}[P(x_{out}) - P(x_*)] \leq \frac{4m+2}{m(\alpha-7)S}[P(x_0) - P(x_*)] + \frac{L\alpha(\alpha-1)}{2m(\alpha-7)S}\|x_0 - x_*\|^2. \qquad (14)$$

*where $x_{out} = \frac{1}{S}\sum_{s=1}^{S}\tilde{x}^s$.*

*Proof.* Because we know $F(\cdot)$ is $L$-smooth, then we have

$$P(x_{k-1/2}) \leq R(x_{k-1/2}) + F(x_{k-1}) + \langle \tilde{\nabla} f_{i_k}(x_{k-1}), x_{k-1/2} - x_{k-1} \rangle$$

$$+ \frac{L}{2}\|x_{k-1/2} - x_{k-1}\|^2 + \langle \nabla f(x_{k-1}) - \tilde{\nabla} f_{i_k}(x_{k-1}), x_{k-1/2} - x_{k-1} \rangle$$

$$\leq R(x_{k-1/2}) + F(x_{k-1}) + \langle \tilde{\nabla} f_{i_k}(x_{k-1}), x_{k-1/2} - x_{k-1} \rangle + \frac{L}{2}\|x_{k-1/2} - x_{k-1}\|^2$$

$$+ \frac{1}{2L(\alpha-1)}\|\nabla F(x_{k-1}) - \tilde{\nabla} f_{i_k}(x_{k-1})\|^2 + \frac{L(\alpha-1)}{2}\|x_{k-1/2} - x_{k-1}\|^2.$$

The second inequality holds due to Young's inequality with parameter $L(\alpha-1)$, where $\alpha$ is a small constant. Then after taking expectation with respect to the sample $i_k$ and using Lemma 5, we obtain

$$\mathbb{E}[P(x_{k-1/2})] \leq R(x_{k-1/2}) + F(x_{k-1}) + \mathbb{E}\langle \tilde{\nabla} f_{i_k}(x_{k-1}), x_{k-1/2} - x_{k-1} \rangle$$

$$+ \frac{L\alpha}{2}\mathbb{E}\|x_{k-1/2} - x_{k-1}\|^2 + \frac{2}{\alpha-1}[P(x_{k-1}) - P(x_*) + P(\tilde{x}) - P(x_*)].$$

Next, we apply Lemma 1 with $x_{k-1}$, $x_k = x_{k-1/2}$, $u = x_*$, and have

$$\mathbb{E}[P(x_{k-1/2})] \leq R(x_*) + F(x_{k-1}) + \mathbb{E}\langle \tilde{\nabla} f_{i_k}(x_{k-1}), x_* - x_{k-1} \rangle$$

$$+ \frac{L\alpha}{2}(\|x_* - x_{k-1}\|^2 - \mathbb{E}\|x_* - x_{k-1/2}\|^2)$$

$$+ \frac{2}{\alpha-1}[P(x_{k-1}) - P(x_*) + P(\tilde{x}) - P(x_*)]$$

$$\leq R(x_*) + F(x_*) + \frac{L\alpha}{2}(\|x_* - x_{k-1}\|^2 - \mathbb{E}\|x_* - x_{k-1/2}\|^2)$$

$$+ \frac{2}{\alpha-1}[P(x_{k-1}) - P(x_*) + P(\tilde{x}) - P(x_*)].$$

The second inequality holds because $\mathbb{E}[\tilde{\nabla} f_{i_k}(x_{k-1})] = \nabla F(x_{k-1})$ and $F(\cdot)$ is convex. Then we know

$$
\begin{aligned}
\mathbb{E}[P(x_{k-1/2}) - P(x_*)] \leq{} & \frac{2}{\alpha - 1}[P(x_{k-1}) - P(x_*) + P(\tilde{x}) - P(x_*)] \\
& + \frac{L\alpha}{2}(\|x_* - x_{k-1}\|^2 - \mathbb{E}\|x_* - x_{k-1/2}\|^2).
\end{aligned}
\tag{15}
$$

And for $P(x_k)$, we can deduce by the same way, and obtain the similar result:

$$
\begin{aligned}
\mathbb{E}[P(x_k) - P(x_*)] \leq{} & \frac{2}{\alpha - 1}[P(x_{k-1/2}) - P(x_*) + P(\tilde{x}) - P(x_*)] \\
& + \frac{L\alpha}{2}(\|x_* - x_{k-1/2}\|^2 - \mathbb{E}\|x_* - x_k\|^2).
\end{aligned}
\tag{16}
$$

We assume that $\alpha$ is sufficiently large to make $\frac{2}{\alpha-1} \leq 1$, and sum (15) and (16) together, then we obtain

$$
\begin{aligned}
\mathbb{E}[P(x_k) - P(x_*)] \leq{} & \frac{2}{\alpha - 1}[P(x_{k-1}) - P(x_*)] + \frac{4}{\alpha - 1}[P(\tilde{x}) - P(x_*)] \\
& + \frac{L\alpha}{2}(\|x_* - x_{k-1}\|^2 - \mathbb{E}\|x_* - x_k\|^2).
\end{aligned}
$$

which is equivalent to

$$
\begin{aligned}
(1 - \frac{2}{\alpha - 1})\mathbb{E}[P(x_k) - P(x_*)] \leq{} & \frac{2}{\alpha - 1}\{[P(x_{k-1}) - P(x_*)] - \mathbb{E}[P(x_k) - P(x_*)]\} \\
& + \frac{4}{\alpha - 1}[P(\tilde{x}) - P(x_*)] + \frac{L\alpha}{2}(\|x_* - x_{k-1}\|^2 - \mathbb{E}\|x_* - x_k\|^2).
\end{aligned}
$$

By summing the previous inequality over $k = 1, ..., m$, we obtain

$$
\begin{aligned}
(1 - \frac{2}{\alpha - 1})\sum_{k=1}^{m}\mathbb{E}[P(x_k) - P(x_*)] \leq{} & \frac{2}{\alpha - 1}\{[P(x_0) - P(x_*)] - [P(x_m) - P(x_*)]\} \\
& + \frac{4m}{\alpha - 1}[P(\tilde{x}) - P(x_*)] + \frac{L\alpha}{2}(\|x_* - x_0\|^2 - \|x_* - x_m\|^2).
\end{aligned}
$$

Since we set $\tilde{x}^s = \frac{1}{m}\sum_{k=1}^{m} x_k$ and $x_0^{s+1} = x_m$, and we know $F(\cdot)$ is a convex function. Thus, we have

$$
\begin{aligned}
(1 - \frac{2}{\alpha - 1})\mathbb{E}[P(\tilde{x}^s) - P(x_*)] \leq{} & \frac{2}{m(\alpha - 1)}\{[P(x_0^s) - P(x_*)] - [P(x_0^{s+1}) - P(x_*)]\} \\
& + \frac{4}{\alpha - 1}[P(\tilde{x}^{s-1}) - P(x_*)] + \frac{L\alpha}{2m}(\|x_* - x_0^s\|^2 - \|x_* - x_0^{s+1}\|^2).
\end{aligned}
$$

By summing the previous inequality over $s = 1, ..., S$, we obtain

$$
\begin{aligned}
(1 - \frac{2}{\alpha - 1})\sum_{s=1}^{S}\mathbb{E}[P(\tilde{x}^s) - P(x_*)] \leq{} & \frac{4}{\alpha - 1}\sum_{s=0}^{S-1}[P(\tilde{x}^s) - P(x_*)] + \frac{L\alpha}{2m}(\|x_* - x_0^1\|^2 - \|x_* - x_m^S\|^2) \\
& + \frac{2}{m(\alpha - 1)}\{[P(x_0^1) - P(x_*)] - [P(x_m^S) - P(x_*)]\}.
\end{aligned}
$$

That is,

$$(1 - \frac{2}{\alpha - 1} - \frac{4}{\alpha - 1}) \sum_{s=1}^{S} \mathbb{E}[P(\tilde{x}^s) - P(x_*)]$$

$$\leq \frac{4}{\alpha - 1}[P(\tilde{x}^0) - P(x_*)] + \frac{2}{m(\alpha - 1)}([P(x_0^1) - P(x_*)] - [P(x_m^S) - P(x_*)])$$

$$+ \frac{L\alpha}{2m}(\|x_* - x_0^1\|^2 - \|x_* - x_m^S\|^2)$$

$$\leq \frac{2}{m(\alpha - 1)}[P(x_0^1) - P(x_*)] + \frac{4}{\alpha - 1}[P(\tilde{x}^0) - P(x_*)] + \frac{L\alpha}{2m}\|x_* - x_0^1\|^2.$$

The first inequality holds due to $1 - \frac{2}{\alpha-1} \geq 1 - \frac{2}{\alpha-1} - \frac{4}{\alpha-1}$ and the second inequality holds because

$P(x_m^S) - P(x_*) \geq 0$ and $\|x_* - x_m^S\|^2 \geq 0$. Because $x_0^1 = \tilde{x}^0$, we have

$$(1 - \frac{6}{\alpha - 1}) \sum_{s=1}^{S} \mathbb{E}[P(\tilde{x}^s) - P(x_*)] \leq (\frac{2}{m(\alpha - 1)} + \frac{4}{\alpha - 1})[P(\tilde{x}^0) - P(x_*)] + \frac{L\alpha}{2m}\|\tilde{x}^0 - x_*\|^2$$

Due to the convexity of $F(\cdot)$, we have

$$\mathbb{E}P(\sum_{s=1}^{S} \tilde{x}^s) - P(x_*) \leq \frac{1}{S} \sum_{s=1}^{S} \mathbb{E}[P(\tilde{x}^s) - P(x_*)]$$

$$\leq \frac{4m + 2}{m(\alpha - 7)S}[P(\tilde{x}^0) - P(x_*)] + \frac{L\alpha(\alpha - 1)}{2m(\alpha - 7)S}\|\tilde{x}^0 - x_*\|^2$$

We note that $\tilde{x}^0 = x_0$, so we have

$$\mathbb{E}P(\sum_{s=1}^{S} \tilde{x}^s) - P(x_*) \leq \frac{4m + 2}{m(\alpha - 7)S}[P(x_0) - P(x_*)] + \frac{L\alpha(\alpha - 1)}{2m(\alpha - 7)S}\|x_0 - x_*\|^2$$

$\square$

# E    MORE EXPERIMENTAL RESULTS

## E.1    THE INFORMATION OF DATA SETS

Table 2: Summary of Data Sets and Regularization Coefficient

| Data sets | Sizes $n$ | Dimensions $d$ | Sparsity | $\lambda_1$ | $\lambda_2$ |
|-----------|-----------|----------------|----------|-------------|-------------|
| a9a | 32,562 | 123 | Sparse | $10^{-6}$ | $10^{-4}$ |
| Covtype | 581,012 | 54 | Dense | $10^{-5}$ | $10^{-8}$ |
| rcv1 | 20,242 | 47,236 | Sparse | $10^{-8}$ | $10^{-10}$ |
| real-sim | 72,309 | 20,598 | Sparse | $10^{-6}$ | $10^{-8}$ |

## E.2    THE PROBLEM MODELS

In this part, we introduce two common problem models. The first one is

$$\min_{x \in \mathbb{R}^d} \frac{1}{2n} \sum_{i=1}^{n} (a_i^T x - b_i)^2 + \lambda_1 \|x\|_1 + \frac{\lambda_2}{2}\|x\|^2.$$

When $\lambda_1 \geq 0$, $\lambda_2 \equiv 0$, we can obtain the Lasso problem, and when $\lambda_1$, $\lambda_2 \geq 0$, we can obtain the Elastic-Net problem, which are all non-smooth optimization problems. The second problem model is

$$\min_{x \in \mathbb{R}^d} \frac{1}{n} \sum_{i=1}^{n} \log(1 + \exp(-b_i x^T a_i)) + \lambda \|x\|_1,$$

which is called the $\ell_1$ norm regularized logistic regression problem.

### E.3 MORE EXPERIMENTAL RESULTS

For more comprehensive comparison, we provide the performance comparison of more algorithms, including SVRG++, MiG and our VR-SExtraGD.

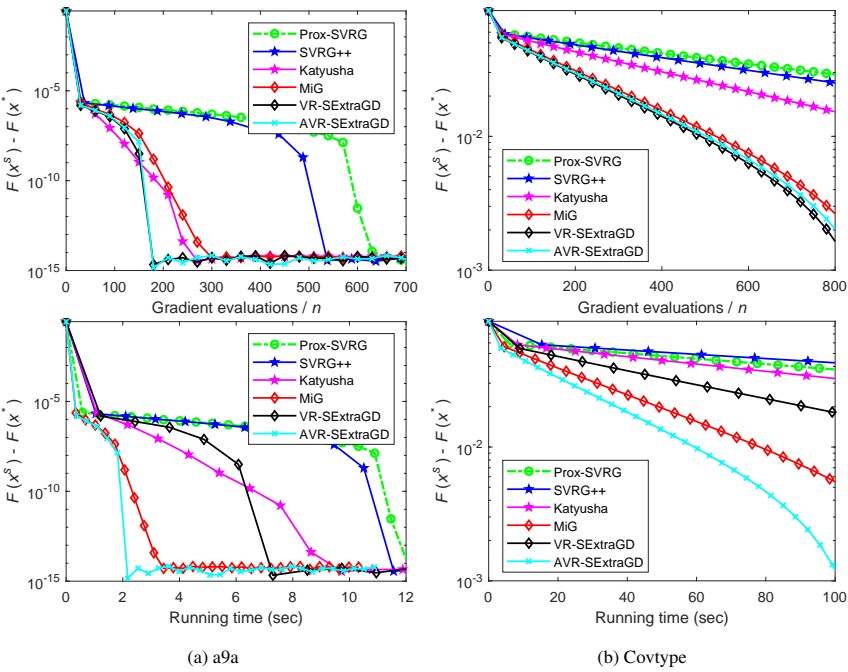

Figure 4: Comparison of experimental results of different algorithms for Lasso on different data sets. The $y$-axis represents the gap between the objective value and the minimum, and the $x$-axis corresponds to the number of effective passes (top) or running time (bottom).

Figure 4 shows the experimental result of different algorithms on different data set to solve the Lasso problem. We can see that AVR-SExtraGD is superior to other algorithms in terms of the number of effective passes and running time. Besides, we note that VR-SExtraGD achieves almost the same result as AVR-SExtraGD in terms of effective passes, which may be due to the advantage of the extragradient structure. But, since VR-SExtraGD needs to calculate the stochastic gradient twice in each inner-iteration, the result in terms of running time is not as good as AVR-SExtraGD.

Moreover, in order to avoid the particularity of the problems solved by our algorithm, we also provide the performance comparison of different algorithms on different data set to solve the $\ell_1$-norm regularized logistic regression problem, as shown in Figure 5.

From Figure 5, we know that our AVR-SExtraGD outperforms other compared algorithms in terms of both effective passes and running time. Although Katyusha is better than our algorithm in terms of effective passes, its result in terms of running time in not as good as ours because of the complicated structure of the algorithm.

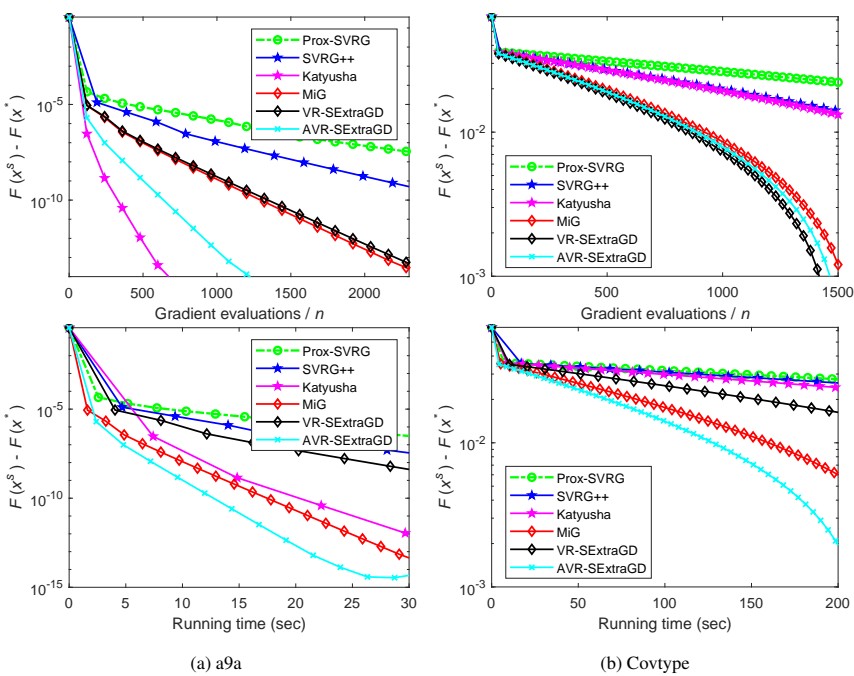

(a) a9a

(b) Covtype

Figure 5: Comparison of experimental results of different algorithms for the $\ell_1$-norm regularized logistic regression problem. on different data sets. The $y$-axis represents the gap between the objective value and the minimum, and the $x$-axis corresponds to the number of effective passes (top) or running time (bottom).

