# OpenReview forum: "Accelerated Variance Reduced Stochastic Extragradient Method for Sparse Machine Learning Problems"
_ICLR.cc/2020/Conference — Reject_

### Official Review · AnonReviewer1 · 2019-10-21
**Official Blind Review #1**

**Rating:** 8

**Review:**

The paper proposes an optimization method for solving unconstrained convex optimization problems where the objective function consists of a sum of several smooth components f_i and a (not necessarily smooth) convex function R. The proposed method AVR-SExtraGD is a stochastic descent method building on the previous algorithms Prox-SVRG (Lin 2014) and Katyusha (Zeyuan 2017). The previous Prox-SVRG method using a proximal operator is explained to converge fast but leads to inaccurate final solutions, while the Katyusha method is an algorithm based on  momentum acceleration. The current paper builds on these two approaches and applies the momentum acceleration technique in a stochastic extragradient descent framework to achieve fast convergence.

I am not working in the field of optimization, therefore, unfortunately I am not in a position to give detailed technical comments for the authors. However, as far as I could follow the paper, it seemed sound and well-written to me in general. I hope the following minor comments may be useful for improving the paper:

- The paper gives detailed explanations about previous work. However, the proposed AVR-SExtraGD algorithm is only presented in the form of a pseudocode in Algorithm 1 and it is not explained in much detail. It would be good to explain and discuss intuitively the steps of the proposed algorithm in the main body of the paper as well, so that it is well understood.

- Algorithm 1 has a set K as input, according to which the solution is updated. How should this set be chosen in practice?

**Experience Assessment:**

I do not know much about this area.

**Review Assessment: Checking Correctness Of Derivations And Theory:**

I did not assess the derivations or theory.

**Review Assessment: Checking Correctness Of Experiments:**

I assessed the sensibility of the experiments.

**Review Assessment: Thoroughness In Paper Reading:**

I made a quick assessment of this paper.

---

> ### Author Response · Authors · 2019-11-15
> **Responses to "Official Blind Review #1**
>
> Question 1: It would be good and well understood to explain intuitively the steps of the proposed algorithm.
>
> Response: Thank you for your valuable suggestion. To address your concern, we have added some explanations for the proposed algorithm in Section 3.1 in the revised manuscript and made it easier to understand.
>
> Question 2: How should “$K$” be chosen in practice?
>
> Response: In our experiments, we used the once extragradient update every 25 inner-iterations, which have also been explained in Section 5 in the manuscript. In practical problems, the frequency of using the extragradient update step can be adjusted according to specific problems and specific requirements. In this way, $K$ can be determined. In addition, we have added more experimental results for choosing the parameter $K$, in the revised manuscript, and also provided a discussion in choosing the parameter.

---

### Official Review · AnonReviewer3 · 2019-10-24
**Official Blind Review #3**

**Rating:** 1

**Review:**

This paper proposed a new stochastic algorithm: AVR-ExtraGD. AVR-ExtraGD combines the extended extragradient method proposed in [3] and the accelerated SVRG method in [1][2]. In their experiments, AVR-ExtraGD outperforms [2] in running time for sparse linear regression.

This paper presents their convergence analysis using results in [1][2]. They showed that the proposed algorithm can achieve O(sqrt{kappa n} log (1 / \epsilon)) complexity for strongly convex problem and O(1/sqrt{epsilon}) for convex problem, which are the best results for both cases.

The idea of an accelerated version of variance reduced stochastic extragradient method is novel. However, there are some issues that the authors should address in order for the paper to match the quality of ICLR.

Each step of extragradient approximates the proximal operator x_{k+1} = argmin_x P(x) + 1/(2 eta_k)\|x – x_k\|_2^2, therefore we would expect a faster and more stable convergence from this method. This paper claims that extragradient reduces the gap between the obtained optimal value and the real optimal value, which is confusing. The update of extragradient is actually biased towards x_k. The claim is then discussed in section 3.1 and 3.2 but is not clearly explained. Besides, how this claim is reflected in the convergence result is not discussed. I encourage the authors to clearly elaborate this claim and make relevant remarks after the main theorems.

To better understand the convergence result, it is important to know how extragradient affects the complexity and choice of hyperparameters such as K, eta_1, eta_2, and beta. Such discussion is not in this paper. I suggest the authors to make these aspects clear.

The experiments compare the proposed algorithm with other algorithms by their running time for lasso and elastic-net. The comparisons show the efficiency of the proposed algorithm. However, a more careful experimental design is required to better demonstrate the performance:
1.	For the choice of inner iterations, choosing m=2n for Katyusha actually requires calculating 5n stochastic gradient because each iteration of Katyusha does gradient updates twice.
2.	This paper only presents comparisons of running time. I encourage the authors to also plots comparisons on number of iterations, which will help revealing where the speed up of AVR-ExtraG comes from.
3.	It is also preferable that the author compare with MiG [1], since the proposed algorithm is an extragradient version of [1].
4.	Please at least solve two different optimization programs (e.g. logistic regression, neural network) so any conclusions are not specific to the oddities of a particular program.

The presentation and structure of this paper need to be improved. Here are some suggestions:
1.	 In Section 1, only provide a high-level literature review and then motivate the work. A comprehensive review can come after the introduction.
2.	In Section 4, put all the lemmas into the appendix while giving more intuitions and remarks.
3.	Issues including notions without pre-definition or reference, typos, and incorrect gramma need to be fixed.


Detailed comments:
1.	From the title, the main application of this work is sparse learning problem. However, how the proposed algorithm benefits sparsity is not discussed. Besides, I suggest the authors to move the asynchronous algorithm in the appendix to the main paper.
2.	The paragraph before section 1.1: lasso and elastic-net are used without citation.
3.	Section 1.1: PGD and SGD are used without citation.
4.	Beginning of page 2: “And” should be “Besides”
5.	“Besides, for accelerating the algorithm and …”: “for accelerating” should be “to accelerate”
6.	Section 1.2: “Nguyen et al. (2017) proposed the idea of extragradient which can be seen as a guide during the process, and introduced it into the optimization problems.” What does “the process” and “it” refers to is unclear.
7.	Section 1.2: the claim extragradient examines the geometry and curvature of the problem is confusing. The geometry of the problem is inspected through a line search step in [3]. However, line search is not discussed in this paper.
8.	Section 1.2: “reduce the gap between the optimal value we get and the real optimal value”, these two kinds optimal values are important notions of this paper but they are not defined.
9.	In Assumption 2, you can refer to Part 2, Section 7 of [5] for the definition of semi-continuity.
10.	“dw is the gradient of the function at w”, what does “the function” refers to?
11.	“APG and Acc-Prox-SVRG” needs citation.
12.	“was proposed to simply the structure of Katyusha”, “simply” should be “simplify”
13.	Section 3.1: “updated with the update rules of MiG”: “with” should be “by”
14.	In the equations of Section 3.2, the equivalent of gradient norm square and function f is incorrect, and the purpose of this equation is unclear.
15.	Section 4.1 Theorem 1: The inequality in theorem 1 is not intuitively related to the convergence rate. I suggest the author to simplify the inequality (For example, Theorem 2.1 in [2]).
16.	The references are not in a uniform format. Conference/Journal names are missing for some references.
17.	One useful reference for this paper is [4], it discussed extragradient for online convex learning.

Additional question:
[5] update the extragradient step by sampling a new stochastic gradient while in this paper the same sample is used twice. How you compare these two approaches in terms of their performance and convergence?

[1] A simple stochastic variance reduced algorithm with fast convergence rates, Zhou et al., 2018.
[2] Katyusha: the first direct acceleration of stochastic gradient methods, Z. Allen-Zhu, 2017
[3] Extragradient method in optimization: Convergence and complexity, T. Nguyen et al., 2017
[4] Online Optimization with Gradual Variations, Chiang et al., 2012
[5] Convex Analysis, R. Rockafella, 1970
[6] Reducing Noise in GAN Training with Variance Reduced Extragradient, Chavdarova et al.  2019


**Experience Assessment:**

I have read many papers in this area.

**Review Assessment: Checking Correctness Of Derivations And Theory:**

I assessed the sensibility of the derivations and theory.

**Review Assessment: Checking Correctness Of Experiments:**

I assessed the sensibility of the experiments.

**Review Assessment: Thoroughness In Paper Reading:**

I read the paper at least twice and used my best judgement in assessing the paper.

---

> ### Author Response · Authors · 2019-11-15
> **Responses to "Official Blind Review #3"**
>
> Thank you for your valuable comments. Through your comments, we get a lot of inspiration and directions to improve the manuscript better.
>
> Question 1: This paper claims that extragradient reduces the gap of the optimal value, which is confusing.
> Response: To address your concern, we have revised the description about the above claim according to the obtained results. We find that the reason why the extragradient can reduce the gap of the optimal value can be explained as follows:
> According to [R1], we know that EEG [R1] can make use of the curvature information of the objective function. Although we change EEG into a stochastic version, the advantage of EEG is still retained to some degree. Thus, we find that the extragradient method is not to directly reduce the gap of the optimal value, but to improve the bad result brought by the gap, which can be verified by our new experiments in the revised manuscript. We have added the above discussion in Section 3 in the revised manuscript.
>
> Question 2: Besides, how this claim is reflected in the convergence result is not discussed. Section 1.2: the claim extragradient examines the geometry and curvature of the problem is confusing.
> Response: We know that the original EEG method is a deterministic algorithm, while our algorithm is a stochastic algorithm. Therefore, the framework and process of theoretical analysis in these two kinds of algorithms are very different.
> In our analysis, due to the introduction of extragradient, there are several issues that are not easy to address. For instance, most of the previous related algorithms only update once in each inner iteration, and thus there is only the relationship between $x_{k-1}$ and $x_k$. In this way, we can use the existing theories and ideas of some algorithms for analysis. But in our algorithm, we update twice in each inner iteration, and thus there are the relationship between $x_{k-1}$ and $x_{k-1/2}$ and the relationship between $x_{k-1/2}$ and $x_k$. Therefore, we need to establish the intermediate connection so that we can get the relationship between $x_{k-1}$ and $x_k$, and finally get the convergence result of the algorithm.
> [R1] uses a linear search method, while it is not applicable for stochastic algorithms, because the direction used in each update of stochastic algorithms is not necessarily the descent direction, and thus the function value may not be reduced after linear search.
>
> Question 3: How does extragradient affect the complexity and choice of hyperparameters such as $K$, $\eta_1$, $\eta_2$, and $\beta$?
> Response: We note that the hyperparameters related to the extragradient are mainly two step sizes, i.e., $\eta_1$ and $\eta_2$. They need to satisfy the conditions given in the manuscript, which are based on EEG step sizes given in [R1]. As for $K$, it is only used to control the frequency of using extragradient, and thus it can be adjusted manually according to specific problems and requirements. For the acceleration hyperparameter $\beta$, it is mainly related to momentum. And its setting can refer to the settings of related algorithms (such as MiG [R2]). In addition, it can also be adjusted manually according to the experimental results. All the discussions have been added in the revised manuscript.
>
> Question 4: A more careful experimental design is required to better demonstrate the performance:
> 1.	The choice of inner iterations.
> 2.	The comparisons on number of iterations.
> 3.	The comparison with MiG [R2].
> 4.	At least solve two different optimization programs (e.g. logistic regression, neural network).
> Response: We note that Katyusha has two options for $y_{k+1}$, as shown in [R3] . When we choose the first option, we need to set $m\!=\!n$ in our paper, but when we choose the second option, $m\!=\!2n$ is appropriate. Besides, in order to make a more comprehensive comparison, we have added more new experiments in the revised manuscript, including the comparison based on iteration, the comparison with more compared algorithms, and the performance comparison when solving logistic regression with $\ell_1$ norm. We also give some reasonable explanations for the experimental results.
>
> Question 5: The presentation and structure of this paper need to be improved.
> Response: We have carefully revised the manuscript according to your suggestions. We have moved all the lemmas to Appendix, fixed typos and incorrect grammar, and added the definitions of some undefined symbols.
>
> [R1] T. Nguyen et al., Extragradient method in optimization: Convergence and complexity, 2017.
> [R2] Zhou et al., A simple stochastic variance reduced algorithm with fast convergence rates, 2018.
> [R3] Allen-Zhu Zeyuan. Katyusha: the first direct acceleration of stochastic gradient methods. 2017.

---

> ### Author Response · Authors · 2019-11-15
> **Responses to "Detailed comments" and "Additional question" of "Official Blind Review #3"**
>
> Response for “Detailed comments”:
> Thank you for checking the manuscript so carefully. According to your “Detailed comments”, we have revised and improved our paper. Besides, here are the explanations of some main questions:
> Response for “1.”: Our algorithm is applicable to both sparse and dense data sets. But in practical problems, the data is usually large-scale and sparse, and thus our main purpose of this algorithm is to solve the problem of large-scale sparse data sets. Besides, our sparse asynchronous variant can take advantage of the sparsity of data, and thus we have moved the asynchronous algorithm to the main body of the paper according to your suggestion.
> Response for “6”: Here “the process” refers to the whole process of solving the problem to be solved. In the optimization problem, it refers to the process of solving the optimal value of the objective function. And “it” refers to the idea of extragradient.
> Response for “10”: Here “the function” means the objective function to be optimized. We have also revised it in the revised manuscript.
> Response for “11”: We have given the citations where “APG” and “Acc-Prox-SVRG” appear in the revised manuscript. Due to the limited space, we do not give the citations where they reappear.
> Response for “14”: The equations of Section 3.2 solve the minimum point of the function, and thus adding and reducing a constant term which is independent of the variable in the function will not change the final minimum point. In addition, the purpose of this equation is to explain that the proximal operator will introduce a gap of the optimal value. In particular, we can add the statement in the revised manuscript.
>
> Additional question:
> [R4] update the extragradient step by sampling a new stochastic gradient while in this paper the same sample is used twice. How you compare these two approaches in terms of their performance and convergence?
> Response: We know that the algorithm without extragradient, such as Prox-SVRG, will sample one stochastic gradient to update in each inner iteration. Therefore, sampling a new stochastic sample to update the extragradient is equivalent to using the algorithm without extragradient to update twice. In this way, we cannot take advantage of the advantage of extragradient. However, we update twice on the same sample point in our algorithm. It is different from the algorithms without extragradient, and can get better results, which can be seen from the experimental comparison between VR-SExtraGD and Prox-SVRG.
>
> [R4] R. Rockafella, Convex Analysis, 1970.

---

### Official Review · AnonReviewer2 · 2019-10-24
**Official Blind Review #2**

**Rating:** 6

**Review:**

This is an optimization algorithm paper, using the idea of "extragradient" and proposing to combine acceleration with proximal gradient descent-type algorithms (Prox-SVRG). Their proposed algorithm, i.e., accelerated variance reduced stochastic extra gradient descent, combines the advantages of Prox-SVRG and momentum acceleration techniques. The authors prove the convergence rate and oracle complexity of their algorithm for strongly convex and non-strongly convex problems. Their experiments on face recognition show improvement on top of Prox-SVRG as well Katyusha. They also propose an asynchronous variant of their algorithm and show that it outperforms other asynchronous baselines.

- technically sound, seems like a nice addition to the variance reduced gradient-type methods. Combines the nice properties of proximal methods with variance reduced gradient-descent.
- Nice summary of recent progress in this research area.
- How does it compare to SVRG++? How about the Proximal Proximal Gradient?
- algorithm suitable for non-smooth optimization problems
- their experimental results look convincing.

Having said that, it seems to me that combining momentum with an existing algorithm is not extremely novel -- I would defer to reviewers who are experts in the optimization area to fully assess the novelty and technical difficulty of the proposed solution.



**Experience Assessment:**

I have published one or two papers in this area.

**Review Assessment: Checking Correctness Of Derivations And Theory:**

I assessed the sensibility of the derivations and theory.

**Review Assessment: Checking Correctness Of Experiments:**

I assessed the sensibility of the experiments.

**Review Assessment: Thoroughness In Paper Reading:**

I read the paper at least twice and used my best judgement in assessing the paper.

---

> ### Author Response · Authors · 2019-11-15
> **Responses to "Official Blind Review #2"**
>
> Question 1: How does it compare to SVRG++ and Proximal Proximal Gradient?
>
> Response: Thank you for the suggestion. To address your concern, we have added some experimental results to compare SVRG++ with the proposed algorithm. All the results show that SVRG++ slightly outperforms Prox-SVRG. However, our AVR-SExtraGD has better performance than SVRG++, due to extragradient and momentum acceleration. As for the Proximal-Proximal-Gradient (PPG) method, when solving the same problem as ours, it is actually proximal gradient method, which means it is a more general algorithm of proximal gradient method. Thus, the comparison with PPG is meaningless.
>
> Question 2: Combining momentum with an existing algorithm is not extremely novel.
>
> Response: In this paper, we first introduced the idea of extragradient descent into Prox-SVRG and propose a new algorithm, called VR-SExtraGD. In particular, we also proposed a new momentum accelerated VR-SExtraGD algorithm, called AVR-SExtraGD. Thus, we not only combine momentum acceleration with the proposed VR-SExtraGD algorithm, but also introduce an innovative idea into the algorithm. Another main contribution of this paper is the convergence results of the proposed algorithms including VR-SExtraGD and AVR-SExtraGD. Due to the introduction of extragtadient descent, our convergence analysis for the proposed algorithms needs more improvement and innovation, which is also our main novelty and overcoming technical difficulty.

---

### Author Response · Authors · 2019-11-15
**Revision uploaded**

We thank all the reviewers for their invaluable comments. We answered all the reviewers' concerns and questions, respectively, and uploaded a revised version of our paper.

---

### Decision · Program_Chairs · 2019-12-19

**Decision:**

Reject

**Comment:**

This paper proposes a stochastic variance reduced extragradient algorithm. The reviewers had a number of concerns which I feel have been adequately addressed by the authors.

That being said, the field of optimizers is crowded and I could not be convinced that the proposed method would be used. In particular, (almost) hyperparameter-free methods are usually preferred (see Adam), which is not the case here.

To be honest, this work is borderline and could have gone either way but was rated lower than other borderline submissions.